# ACT: Asymptotic Conditional Transport

## Abstract

We propose conditional transport (CT) as a new divergence to measure the difference between two probability distributions. The CT divergence consists of the expected cost of a forward CT, which constructs a navigator to stochastically transport a data point of one distribution to the other distribution, and that of a backward CT which reverses the transport direction. To apply it to the distributions whose probability density functions are unknown but random samples are accessible, we further introduce asymptotic CT (ACT), whose estimation only requires access to mini-batch based discrete empirical distributions. Equipped with two navigators that amortize the computation of conditional transport plans, the ACT divergence comes with unbiased sample gradients that are straightforward to compute, making it amenable to mini-batch stochastic gradient descent based optimization. When applied to train a generative model, the ACT divergence is shown to strike a good balance between mode covering and seeking behaviors and strongly resist mode collapse. To model high-dimensional data, we show that it is sufficient to modify the adversarial game of an existing generative adversarial network (GAN) to a game played by a generator, a forward navigator, and a backward navigator, which try to minimize a distribution-to-distribution transport cost by optimizing both the distribution of the generator and conditional transport plans specified by the navigators, versus a critic that does the opposite by inflating the point-to-point transport cost. On a wide variety of benchmark datasets for generative modeling, substituting the default statistical distance of an existing GAN with the ACT divergence is shown to consistently improve the performance.

## 1 Introduction

Measuring the difference between two probability distributions is a fundamental problem in statistics and machine learning (Cover, 1999; Bishop, 2006; Murphy, 2012). A variety of statistical distances have been proposed to quantify the difference, which often serves as the first step to build a generative model. Commonly used statistical distances include the Kullback–Leibler (KL) divergence (Kullback and Leibler, 1951), Jensen–Shannon (JS) divergence (Lin, 1991), and Wasserstein distance (Kantorovich, 2006). While being widely used for generative modeling (Kingma and Welling, 2013; Goodfellow et al., 2014; Arjovsky et al., 2017; Balaji et al., 2019), they all have their own limitations. The KL divergence, directly related to both maximum likelihood estimation and variational inference, is amenable to mini-batch stochastic gradient descent (SGD) based optimization (Wainwright and Jordan, 2008; Hoffman et al., 2013; Blei et al., 2017). However, it requires the two probability distributions to share the same support, and hence is often inapplicable if either of them is an implicit distribution whose probability density function (PDF) is unknown (Mohamed and Lakshminarayanan, 2016; Huszár, 2017; Tran et al., 2017; Yin and Zhou, 2018). The JS divergence is directly related to the mini-max loss of a generative adversarial net (GAN) when the discriminator is optimal (Goodfellow et al., 2014). However, it is difficult to maintain a good balance between the generator and discriminator, making GANs notoriously brittle to train. The Wasserstein distance is a widely used metric that allows the two distributions to have non-overlapping supports (Villani, 2008; Santambrogio, 2015; Peyré and Cuturi, 2019). However, it is challenging to estimate in its primal form and generally results in biased sample gradients when its dual form is employed (Arjovsky et al., 2017; Bellemare et al., 2017; Bottou et al., 2017; Bińkowski et al., 2018; Bernton et al., 2019).

To address the limitations of existing measurement methods, we introduce conditional transport (CT) as a new divergence to quantify the difference between two probability distributions. We

refer to them as the source and target distributions and denote their probability density functions (PDFs) as $p_X(\boldsymbol{x})$ and $p_Y(\boldsymbol{y})$, respectively. The CT divergence is defined with a bidirectional distribution-to-distribution transport. It consists of a forward CT that transports the source to target distribution, and a backward CT that reverses the transport direction. Our intuition is that given a source (target) point, it is more likely to be transported to a target (source) point closer to it. Denoting $d(\boldsymbol{x}, \boldsymbol{y}) = d(\boldsymbol{y}, \boldsymbol{x})$ as a learnable function and $c(\boldsymbol{x}, \boldsymbol{y}) = c(\boldsymbol{y}, \boldsymbol{x}) \geq 0$, where the equality is true when $\boldsymbol{x} = \boldsymbol{y}$, as the point-to-point transport cost, the goal is to minimize the transport cost between two distributions. The forward CT is constructed in three steps: 1) We define a forward "navigator" as $\pi(\boldsymbol{y} \,|\, \boldsymbol{x}) = e^{-d(\boldsymbol{x}, \boldsymbol{y})} p_Y(\boldsymbol{y}) / \int e^{-d(\boldsymbol{x}, \boldsymbol{y})} p_Y(\boldsymbol{y}) \mathrm{d}\boldsymbol{y}$, a conditional distribution specifying how likely a given source point $\boldsymbol{x}$ will be transported to distribution $p_Y(\boldsymbol{y})$ via path $\boldsymbol{x} \rightarrow \boldsymbol{y}$; 2) We define the cost of a forward $\boldsymbol{x}$-transporting CT as $\int c(\boldsymbol{x}, \boldsymbol{y}) \pi(\boldsymbol{y} \,|\, \boldsymbol{x}) \mathrm{d}\boldsymbol{y}$, the expected cost of employing the forward navigator to transport $\boldsymbol{x}$ to a random target point; 3) We define the total cost of the forward CT as $\int p_X(\boldsymbol{x}) \int c(\boldsymbol{x}, \boldsymbol{y}) \pi(\boldsymbol{y} \,|\, \boldsymbol{x}) \mathrm{d}\boldsymbol{y} \mathrm{d}\boldsymbol{x}$, which is the expectation of the cost of a forward $\boldsymbol{x}$-transporting CT with respect to $p_X(\boldsymbol{x})$. Similarly, we construct the backward CT by first defining a backward navigator as $\pi(\boldsymbol{x} \,|\, \boldsymbol{y}) = e^{-d(\boldsymbol{x}, \boldsymbol{y})} p_X(\boldsymbol{x}) / \int e^{-d(\boldsymbol{x}, \boldsymbol{y})} p_X(\boldsymbol{x}) \mathrm{d}\boldsymbol{x}$ and then its total cost as $\int p_Y(\boldsymbol{y}) \int c(\boldsymbol{x}, \boldsymbol{y}) \pi(\boldsymbol{x} \,|\, \boldsymbol{y}) \mathrm{d}\boldsymbol{x} \mathrm{d}\boldsymbol{y}$. Estimating the CT divergence involves both $\pi(\boldsymbol{x} \,|\, \boldsymbol{y})$ and $\pi(\boldsymbol{y} \,|\, \boldsymbol{x})$, which, however, are generally intractable to evaluate and sample from, except for a few limited settings where both $p_X(\boldsymbol{x})$ and $p_Y(\boldsymbol{y})$ are exponential family distributions conjugate to $e^{-d(\boldsymbol{x}, \boldsymbol{y})}$.

To apply the CT divergence in a general setting where we only have access to random samples from the distributions, we introduce asymptotic CT (ACT) as a divergence measure that is friendly to mini-batch SGD based optimization. The ACT divergence is the expected value of the CT divergence, whose $p_X(\boldsymbol{x})$ and $p_Y(\boldsymbol{y})$ are both replaced with their discrete empirical distributions, respectively supported on $N$ independent, and identically distributed ($iid$) random samples from $p_X(\boldsymbol{x})$ and $M$ $iid$ random samples from $p_Y(\boldsymbol{y})$. The ACT divergence is asymptotically equivalent to CT divergence when both $N \rightarrow \infty$ and $M \rightarrow \infty$. Intuitively, it can also be interpreted as performing both a forward one-to-$M$ stochastic CT from the source to target and a backward one-to-$N$ stochastic CT from the target to source, with the expected cost providing an unbiased sample estimate of the ACT divergence.

We show that similar to the KL divergence, ACT provides unbiased sample gradients, but different from it, neither $p_X(\boldsymbol{x})$ nor $p_Y(\boldsymbol{y})$ needs to be known. Similar to the Wasserstein distance, it does not require the distributions to share the same support, but different from it, the sample estimates of ACT and its gradients are unbiased and straightforward to compute. In GANs or Wasserstein GANs (Arjovsky et al., 2017), having an optimal discriminator or critic is required to unbiasedly estimate the JS divergence or Wasserstein distance and hence the gradients of the generator (Bottou et al., 2017). However, this is rarely the case in practice, motivating a common remedy to stabilize the training by carefully regularizing the gradients, such as clipping or normalizing their values (Gulrajani et al., 2017; Miyato et al., 2018). By contrast, in an adversarial game under ACT, the optimization of the critic, which manipulates the point-to-point transport cost $c(\boldsymbol{x}, \boldsymbol{y})$ but not the navigators' conditional distributions for $\boldsymbol{x} \rightarrow \boldsymbol{y}$ and $\boldsymbol{x} \leftarrow \boldsymbol{y}$, has no impact on how ACT is estimated. For this reason, the sample gradients stay unbiased regardless of how well the critic is optimized.

To demonstrate the use of the ACT (or CT) divergence, we apply it to train implicit (or explicit) distributions to model both 1D and 2D toy data, MNIST digits, and natural images. The implicit distribution is defined by a deep generative model (DGM) that is simple to sample from. We focus on adapting existing GANs, with minimal changes to their settings except for substituting the statistical distances in their loss functions with the ACT divergence. We leave tailoring the network architectures to the ACT divergence to future study. More specifically, we modify the GAN loss function to an adversarial game between a generator, a forward navigator, and a backward navigator, which try to minimize the distribution-to-distribution transport cost by optimizing both the fake data distribution $p_Y(\boldsymbol{y})$ and two conditional point-to-point navigation-path distributions $\pi(\boldsymbol{y} \,|\, \boldsymbol{x})$ and $\pi(\boldsymbol{x} \,|\, \boldsymbol{y})$, versus a critic that does the opposite by inflating the point-to-point transport cost $c(\boldsymbol{x}, \boldsymbol{y})$. Modifying an existing (Wasserstein) GAN with the ACT divergence, our experiments show consistent improvements in not only quantitative performance and generation quality, but also learning stability.

## 2    Conditional Transport with Generator, Navigators, and Critic

Denote $\boldsymbol{x}$ as a data taking its value in $\mathbb{R}^V$. In practice, we observe a finite set $\mathcal{X} = \{\boldsymbol{x}_i\}_{i=1}^{|\mathcal{X}|}$, consisting of $|\mathcal{X}|$ data samples assumed to be $iid$ drawn from $p_X(\boldsymbol{x})$. Given $\mathcal{X}$, the usual task is to learn a distribution to approximate $p_X(\boldsymbol{x})$, explaining how the data in $\mathcal{X}$ are generated. To approximate

$p_X(\boldsymbol{x})$, we consider a DGM defined as $\boldsymbol{y} = G_{\boldsymbol{\theta}}(\boldsymbol{\epsilon})$, $\boldsymbol{\epsilon} \sim p(\boldsymbol{\epsilon})$, where $G_{\boldsymbol{\theta}}$ is a generator that transforms noise $\boldsymbol{\epsilon} \sim p(\boldsymbol{\epsilon})$ via a deep neural network parameterized by $\boldsymbol{\theta}$ to generate random sample $\boldsymbol{y} \in \mathbb{R}^V$. While the PDF of the generator, denoted as $p_{\boldsymbol{\theta}}(\boldsymbol{y})$, is often intractable to evaluate, it is straightforward to draw $\boldsymbol{y} \sim p_{\boldsymbol{\theta}}(\boldsymbol{y})$ with $G_{\boldsymbol{\theta}}$. Denote both $\mu(\mathrm{d}\boldsymbol{x}) = p_X(\boldsymbol{x})\mathrm{d}\boldsymbol{x}$ and $\nu(\mathrm{d}\boldsymbol{y}) = p_{\boldsymbol{\theta}}(\boldsymbol{y})\mathrm{d}\boldsymbol{y}$ as continuous probability measures over $\mathbb{R}^V$, with $\mu(\mathbb{R}^V) = \int_{\mathbb{R}^V} p_X(\boldsymbol{x})\mathrm{d}\boldsymbol{x} = 1$ and $\nu(\mathbb{R}^V) = \int_{\mathbb{R}^V} p_{\boldsymbol{\theta}}(\boldsymbol{y})\mathrm{d}\boldsymbol{y} = 1$. The Wasserstein distance in its primal form can be defined with Kantorovich's optimal transport problem (Kantorovich, 2006; Villani, 2008; Santambrogio, 2015; Peyré and Cuturi, 2019):

$$\mathcal{W}(\mu, \nu) = \min_{\pi \in \Pi(\mu,\nu)}\{\textstyle\int_{\mathbb{R}^V \times \mathbb{R}^V} c(\boldsymbol{x}, \boldsymbol{y})\pi(\mathrm{d}\boldsymbol{x}, \mathrm{d}\boldsymbol{y})\} = \min_{\pi \in \Pi(\mu,\nu)}\{\mathbb{E}_{(\boldsymbol{x},\boldsymbol{y})\sim\pi(\boldsymbol{x},\boldsymbol{y})}[c(\boldsymbol{x}, \boldsymbol{y})]\}, \quad (1)$$

where the minimum is taken over $\Pi(\mu, \nu)$, defined as the set of all possible joint probability measures $\pi$ on $\mathbb{R}^V \times \mathbb{R}^V$, with marginals $\pi(A, \mathbb{R}^V) = \mu(A)$ and $\pi(\mathbb{R}^V, A) = \nu(A)$ for any Borel set $A \subset \mathbb{R}^V$. When $c(\boldsymbol{x}, \boldsymbol{y}) = \|\boldsymbol{x} - \boldsymbol{y}\|$, we obtain the Wasserstein-1 distance, also known as the Earth Mover's distance, for which there exists a dual form according to the Kantorovich duality as

$$\mathcal{W}_1(\mu, \nu) = \sup_{f \in \mathrm{Lip}^1}\{\mathbb{E}_{\boldsymbol{x} \sim p_X(\boldsymbol{x})}[f(\boldsymbol{x})] - \mathbb{E}_{\boldsymbol{y} \sim p_Y(\boldsymbol{y})}[f(\boldsymbol{y})]\},$$

where $f$ is referred to as the "critic" and $\mathrm{Lip}^1$ denotes the set of all 1-Lipschitz functions (Villani, 2008). Intuitively, the critic $f$ plays the role of "amortizing" the computation of the optimal transport plan. However, as it is difficult to ensure the 1-Lipschitz constraint, one often resorts to approximations (Arjovsky et al., 2017; Gulrajani et al., 2017; Wei et al., 2018; Miyato et al., 2018) that inevitably introduce bias into the estimation of $\mathcal{W}_1$ and its gradient (Bellemare et al., 2017; Bottou et al., 2017).

## 2.1 FORWARD AND BACKWARD NAVIGATORS AND CONDITIONAL TRANSPORT PLANS

Constraining $\pi \in \Pi(\mu, \nu)$, the Wasserstein distance satisfies $\mathcal{W}(\mu, \nu) = \mathcal{W}(\nu, \mu)$. By contrast, the proposed divergence allows $\pi \notin \Pi(\mu, \nu)$. Denote $\mathcal{T}_{\boldsymbol{\phi}}(\cdot) \in \mathbb{R}^H$ as a neural network based function, transforming its input in $\mathbb{R}^V$ into a feature vector in $\mathbb{R}^H$, and $d(\boldsymbol{h}_1, \boldsymbol{h}_2)$ as a function that measures the difference between $\boldsymbol{h}_1, \boldsymbol{h}_2 \in \mathbb{R}^H$. We introduce a forward CT, whose transport cost is defined as

$$\mathcal{C}_{\boldsymbol{\phi},\boldsymbol{\theta}}(\mu \rightarrow \nu) = \mathbb{E}_{\boldsymbol{x} \sim p_X(\boldsymbol{x})}\mathbb{E}_{\boldsymbol{y} \sim \pi_{\boldsymbol{\phi}}(\boldsymbol{y}\,|\,\boldsymbol{x})}[c(\boldsymbol{x}, \boldsymbol{y})], \quad \pi_{\boldsymbol{\phi}}(\boldsymbol{y}\,|\,\boldsymbol{x}) \stackrel{\text{def.}}{=} \frac{e^{-d(\mathcal{T}_{\boldsymbol{\phi}}(\boldsymbol{x}),\mathcal{T}_{\boldsymbol{\phi}}(\boldsymbol{y}))}p_{\boldsymbol{\theta}}(\boldsymbol{y})}{\int e^{-d(\mathcal{T}_{\boldsymbol{\phi}}(\boldsymbol{x}),\mathcal{T}_{\boldsymbol{\phi}}(\boldsymbol{y}))}p_{\boldsymbol{\theta}}(\boldsymbol{y})\mathrm{d}\boldsymbol{y}}, \quad (2)$$

where $\pi_{\boldsymbol{\phi}}(\boldsymbol{y}\,|\,\boldsymbol{x})$ will be analogized to the forward navigator that defines the forward conditional transport plan. Similarly, we introduce the backward CT, whose transport cost is defined as

$$\mathcal{C}_{\boldsymbol{\phi},\boldsymbol{\theta}}(\mu \leftarrow \nu) = \mathbb{E}_{\boldsymbol{y} \sim p_{\boldsymbol{\theta}}(\boldsymbol{y})}\mathbb{E}_{\boldsymbol{x} \sim \pi_{\boldsymbol{\phi}}(\boldsymbol{x}\,|\,\boldsymbol{y})}[c(\boldsymbol{x}, \boldsymbol{y})], \quad \pi_{\boldsymbol{\phi}}(\boldsymbol{x}\,|\,\boldsymbol{y}) \stackrel{\text{def.}}{=} \frac{e^{-d(\mathcal{T}_{\boldsymbol{\phi}}(\boldsymbol{x}),\mathcal{T}_{\boldsymbol{\phi}}(\boldsymbol{y}))}p_X(\boldsymbol{x})}{\int e^{-d(\mathcal{T}_{\boldsymbol{\phi}}(\boldsymbol{x}),\mathcal{T}_{\boldsymbol{\phi}}(\boldsymbol{y}))}p_X(\boldsymbol{x})\mathrm{d}\boldsymbol{x}}, \quad (3)$$

where $\pi_{\boldsymbol{\phi}}(\boldsymbol{x}\,|\,\boldsymbol{y})$ will be analogized to a backward navigator. We now define the CT problem as

$$\min_{\boldsymbol{\phi},\boldsymbol{\theta}}\{\mathcal{C}_{\boldsymbol{\phi},\boldsymbol{\theta}}(\mu, \nu)\}, \quad \mathcal{C}_{\boldsymbol{\phi},\boldsymbol{\theta}}(\mu, \nu) \stackrel{\text{def.}}{=} \tfrac{1}{2}\mathcal{C}_{\boldsymbol{\phi},\boldsymbol{\theta}}(\mu \rightarrow \nu) + \tfrac{1}{2}\mathcal{C}_{\boldsymbol{\phi},\boldsymbol{\theta}}(\mu \leftarrow \nu), \quad (4)$$

where $\mathcal{C}_{\boldsymbol{\phi},\boldsymbol{\theta}}(\mu, \nu) = \mathcal{C}_{\boldsymbol{\phi},\boldsymbol{\theta}}(\nu, \mu)$ will be referred to as the CT divergence between $\mu$ and $\nu$.

**Lemma 1.** *If $\boldsymbol{y} \sim p_{\boldsymbol{\theta}}(\boldsymbol{y})$ is equal to $\boldsymbol{x} \sim p_X(\boldsymbol{x})$ in distribution and $\mathcal{T}_{\boldsymbol{\phi}}$ is chosen such that $e^{-d(\mathcal{T}_{\boldsymbol{\phi}}(\boldsymbol{x}),\mathcal{T}_{\boldsymbol{\phi}}(\boldsymbol{y}))} = \mathbf{1}(\boldsymbol{x} = \boldsymbol{y})$, where $\mathbf{1}(\cdot)$ is an indicator function, then both the the joint probability measure $\pi$ defined with $p_X(\boldsymbol{x})\pi_{\boldsymbol{\phi}}(\boldsymbol{y}\,|\,\boldsymbol{x})$ in (2) and that with $p_{\boldsymbol{\theta}}(\boldsymbol{y})\pi_{\boldsymbol{\phi}}(\boldsymbol{x}\,|\,\boldsymbol{y})$ in (3) are in $\Pi(\mu, \nu)$.*

**Lemma 2.** *If $\boldsymbol{y} \sim p_{\boldsymbol{\theta}}(\boldsymbol{y})$ is equal to $\boldsymbol{x} \sim p_X(\boldsymbol{x})$ in distribution, then $C_{\boldsymbol{\phi},\boldsymbol{\theta}}(\mu, \nu) = \mathcal{C}_{\boldsymbol{\phi},\boldsymbol{\theta}}(\mu \rightarrow \nu) = \mathcal{C}_{\boldsymbol{\phi},\boldsymbol{\theta}}(\mu \leftarrow \nu) \geq \mathcal{W}(\mu, \nu) = 0$, where the equality can be achieved if $e^{-d(\mathcal{T}_{\boldsymbol{\phi}}(\boldsymbol{x}),\mathcal{T}_{\boldsymbol{\phi}}(\boldsymbol{y}))} = \mathbf{1}(\boldsymbol{x} = \boldsymbol{y})$.*

The proofs are deferred to Appendix A. Note in general, before both $\boldsymbol{\theta}$ and $\boldsymbol{\phi}$ reach their optimums, the conditions specified in Lemmas 1 and 2 are not satisfied and the joint probability measure $\pi$ defined with $p_X(\boldsymbol{x})\pi_{\boldsymbol{\phi}}(\boldsymbol{y}\,|\,\boldsymbol{x})$ or $p_{\boldsymbol{\theta}}(\boldsymbol{y})\pi_{\boldsymbol{\phi}}(\boldsymbol{x}\,|\,\boldsymbol{y})$ is not restricted to be in $\Pi(\mu, \nu)$, and hence it is possible for $\mathcal{C}_{\boldsymbol{\phi},\boldsymbol{\theta}}(\mu \rightarrow \nu)$, $\mathcal{C}_{\boldsymbol{\phi},\boldsymbol{\theta}}(\mu \leftarrow \nu)$, or $\mathcal{C}_{\boldsymbol{\phi},\boldsymbol{\theta}}(\mu, \nu)$ to go below $\mathcal{W}(\mu, \nu)$ during training.

## 2.2 ASYMPTOTIC CONDITIONAL TRANSPORT

Computing the CT divergence requires either knowing the PDFs of both navigators $\pi_{\boldsymbol{\phi}}(\boldsymbol{y}\,|\,\boldsymbol{x})$ and $\pi_{\boldsymbol{\phi}}(\boldsymbol{x}\,|\,\boldsymbol{y})$, or being able to draw random samples from them. However, usually neither is true unless both $p_{\boldsymbol{\theta}}(\boldsymbol{y})$ and $p_X(\boldsymbol{x})$ are known and conjugate to $e^{-d(\mathcal{T}_{\boldsymbol{\phi}}(\boldsymbol{x}),\mathcal{T}_{\boldsymbol{\phi}}(\boldsymbol{y}))}$. For example, if $d(\mathcal{T}_{\boldsymbol{\phi}}(\boldsymbol{x}), \mathcal{T}_{\boldsymbol{\phi}}(\boldsymbol{y})) = \|\boldsymbol{\phi}\boldsymbol{x} - \boldsymbol{\phi}\boldsymbol{y}\|_2^2 = (\boldsymbol{x} - \boldsymbol{y})^T(\boldsymbol{\phi}^T\boldsymbol{\phi})(\boldsymbol{x} - \boldsymbol{y})$, where $\boldsymbol{\phi} \in \mathbb{R}^{V \times V}$ is a full-rank matrix, and both $p_X(\boldsymbol{x})$ and $p_{\boldsymbol{\theta}}(\boldsymbol{y})$ are multivariate Gaussian distributions, then one may show that both $\pi_{\boldsymbol{\phi}}(\boldsymbol{y}\,|\,\boldsymbol{x})$ and $\pi_{\boldsymbol{\phi}}(\boldsymbol{x}\,|\,\boldsymbol{y})$ are multivariate Gaussian distributions. In the experimental results

section, we will provide a univariate normal distribution based toy examples for illustration. Below we show how to apply the CT divergence in a general setting that only requires access to random samples of both $\boldsymbol{x}$ and $\boldsymbol{y}$. While knowing neither $p_X(\boldsymbol{x})$ nor $p_{\boldsymbol{\theta}}(\boldsymbol{y})$, we can obtain mini-batch based empirical probability measures $\hat{\mu}_N$ and $\hat{\nu}_M$, as defined below, to guide the optimization of $G_{\boldsymbol{\theta}}$ in an iterative manner. With $N$ random observations sampled without replacement from $\mathcal{X}$, we define

$$\hat{\mu}_N = \tfrac{1}{N} \sum_{i=1}^N \delta_{\boldsymbol{x}_i}, \quad \{\boldsymbol{x}_1, \dots, \boldsymbol{x}_N\} \subseteq \mathcal{X} \tag{5}$$

as an empirical probability measure for $\boldsymbol{x}$. Similarly, with $M$ random samples of the generator, we define an empirical probability measure for $\boldsymbol{y}$ as

$$\hat{\nu}_M = \tfrac{1}{M} \sum_{j=1}^M \delta_{\boldsymbol{y}_j}, \quad \boldsymbol{y}_j = G_{\boldsymbol{\theta}}(\boldsymbol{\epsilon}_j), \quad \boldsymbol{\epsilon}_j \overset{iid}{\sim} p(\boldsymbol{\epsilon}) \ . \tag{6}$$

Substituting $p_{\boldsymbol{\theta}}(\boldsymbol{y})$ in (2) with $\hat{\nu}_M(\boldsymbol{y})$, the continuous forward navigator becomes a discrete one as

$$\hat{\pi}_{\boldsymbol{\phi}}(\boldsymbol{y} \,|\, \boldsymbol{x}) = \sum_{j=1}^M \hat{\pi}_M(\boldsymbol{y}_j \,|\, \boldsymbol{x}, \boldsymbol{\phi}) \delta_{\boldsymbol{y}_j}, \quad \hat{\pi}_M(\boldsymbol{y}_j \,|\, \boldsymbol{x}, \boldsymbol{\phi}) \overset{\text{def.}}{=} \frac{e^{-d(\mathcal{T}_{\boldsymbol{\phi}}(\boldsymbol{x}), \mathcal{T}_{\boldsymbol{\phi}}(\boldsymbol{y}_j))}}{\sum_{j'=1}^M e^{-d(\mathcal{T}_{\boldsymbol{\phi}}(\boldsymbol{x}), \mathcal{T}_{\boldsymbol{\phi}}(\boldsymbol{y}_{j'}))}}. \tag{7}$$

Thus the cost of a forward CT becomes

$$\mathcal{C}_{\boldsymbol{\phi},\boldsymbol{\theta}}(\mu \to \hat{\nu}_M) = \mathbb{E}_{\boldsymbol{x} \sim p_X(\boldsymbol{x})} \mathbb{E}_{\boldsymbol{y} \sim \hat{\pi}_{\boldsymbol{\phi}}(\boldsymbol{y} \,|\, \boldsymbol{x})}[c(\boldsymbol{x}, \boldsymbol{y})] = \mathbb{E}_{\boldsymbol{x} \sim p_X(\boldsymbol{x})} \left[ \mathcal{C}_{\boldsymbol{\phi},\boldsymbol{\theta}}(\boldsymbol{x} \to \hat{\nu}_M) \right],$$

$$\text{where} \quad \mathcal{C}_{\boldsymbol{\phi},\boldsymbol{\theta}}(\boldsymbol{x} \to \hat{\nu}_M) \overset{\text{def.}}{=} \sum_{j=1}^M c(\boldsymbol{x}, \boldsymbol{y}_j) \hat{\pi}_M(\boldsymbol{y}_j \,|\, \boldsymbol{x}, \boldsymbol{\phi}). \tag{8}$$

Similarly, we have the backward navigator and the cost of backward CT as

$$\hat{\pi}_{\boldsymbol{\phi}}(\boldsymbol{x} \,|\, \boldsymbol{y}) = \sum_{i=1}^N \hat{\pi}_N(\boldsymbol{x}_i \,|\, \boldsymbol{y}, \boldsymbol{\phi}) \delta_{\boldsymbol{x}_i}, \quad \hat{\pi}_N(\boldsymbol{x}_i \,|\, \boldsymbol{y}, \boldsymbol{\phi}) \overset{\text{def.}}{=} \frac{e^{-d(\mathcal{T}_{\boldsymbol{\phi}}(\boldsymbol{x}_i), \mathcal{T}_{\boldsymbol{\phi}}(\boldsymbol{y}))}}{\sum_{i'=1}^M e^{-d(\mathcal{T}_{\boldsymbol{\phi}}(\boldsymbol{x}_{i'}), \mathcal{T}_{\boldsymbol{\phi}}(\boldsymbol{y}))}}, \tag{9}$$

$$\mathcal{C}_{\boldsymbol{\phi},\boldsymbol{\theta}}(\hat{\mu}_N \leftarrow \nu) = \mathbb{E}_{\boldsymbol{y} \sim p_{\boldsymbol{\theta}}(\boldsymbol{y})} \mathbb{E}_{\boldsymbol{x} \sim \hat{\pi}_{\boldsymbol{\phi}}(\boldsymbol{x} \,|\, \boldsymbol{y})}[c(\boldsymbol{x}, \boldsymbol{y})] = \mathbb{E}_{\boldsymbol{y} \sim p_{\boldsymbol{\theta}}(\boldsymbol{y})} \left[ \mathcal{C}_{\boldsymbol{\phi},\boldsymbol{\theta}}(\hat{\mu}_N \leftarrow \boldsymbol{y}) \right],$$

$$\text{where} \quad \mathcal{C}_{\boldsymbol{\phi},\boldsymbol{\theta}}(\hat{\mu}_N \leftarrow \boldsymbol{y}) \overset{\text{def.}}{=} \sum_{i=1}^N c(\boldsymbol{x}_i, \boldsymbol{y}) \hat{\pi}_N(\boldsymbol{x}_i \,|\, \boldsymbol{y}, \boldsymbol{\phi}). \tag{10}$$

Combining both the forward and backward CTs, we define the asymptotic CT (ACT) problem as

$$\min_{\boldsymbol{\phi},\boldsymbol{\theta}} \{ \mathcal{C}_{\boldsymbol{\phi},\boldsymbol{\theta}}(\mu, \nu, N, M) \}, \tag{11}$$

where $\mathcal{C}_{\boldsymbol{\phi},\boldsymbol{\theta}}(\mu, \nu, N, M)$ is the ACT divergence defined as

$$\mathcal{C}_{\boldsymbol{\phi},\boldsymbol{\theta}}(\mu, \nu, N, M) = \tfrac{1}{2} \mathbb{E}_{\boldsymbol{y}_{1:M} \overset{iid}{\sim} p_{\boldsymbol{\theta}}(\boldsymbol{y})}[\mathcal{C}_{\boldsymbol{\phi},\boldsymbol{\theta}}(\mu \to \hat{\nu}_M)] + \tfrac{1}{2} \mathbb{E}_{\boldsymbol{x}_{1:N} \overset{iid}{\sim} p_X(\boldsymbol{x})}[\mathcal{C}_{\boldsymbol{\phi},\boldsymbol{\theta}}(\hat{\mu}_N \leftarrow \nu)]. \tag{12}$$

**Lemma 3.** *The ACT divergence is asymptotic that* $\lim_{N,M \to \infty} \mathcal{C}_{\boldsymbol{\phi},\boldsymbol{\theta}}(\mu, \nu, N, M) = \mathcal{C}_{\boldsymbol{\phi},\boldsymbol{\theta}}(\mu, \nu)$.

**Lemma 4.** *With* $\boldsymbol{x}_{1:N} \overset{iid}{\sim} p_X(\boldsymbol{x})$ *and* $\boldsymbol{y}_{1:M} \overset{iid}{\sim} p_{\boldsymbol{\theta}}(\boldsymbol{y})$ *and drawing* $\boldsymbol{x} \sim \hat{\mu}_N(\boldsymbol{x})$ *and* $\boldsymbol{y} \sim \hat{\nu}_M(\boldsymbol{y})$, *an unbiased sample estimator of the ACT divergence can be expressed as*

$$\mathcal{L}_{\boldsymbol{\phi},\boldsymbol{\theta}}(\boldsymbol{x}_{1:N}, \boldsymbol{y}_{1:M}) = \tfrac{1}{2} \sum_{j=1}^M c(\boldsymbol{x}, \boldsymbol{y}_j) \hat{\pi}_M(\boldsymbol{y}_j \,|\, \boldsymbol{x}, \boldsymbol{\phi}) + \tfrac{1}{2} \sum_{i=1}^N c(\boldsymbol{x}_i, \boldsymbol{y}) \hat{\pi}_N(\boldsymbol{x}_i \,|\, \boldsymbol{y}, \boldsymbol{\phi}). \tag{13}$$

Intuitively, the first term in the summation can be interpreted as the expected cost of following the forward navigator to stochastically transport a random source point $\boldsymbol{x}$ to one of the $M$ randomly instantiated "anchors" of the target distribution. The second term shares a similar interpretation. Note in optimal transport, the Wasserstein distance $\mathcal{W}(\mu, \nu)$ in its primal form, shown in (1), is in general intractable to compute. To use the primal form, one often resorts to the sample Wasserstein distance defined as $\mathcal{W}(\hat{\mu}_N, \hat{\nu}_M)$, computing which, however, requires solving a combinatorial optimization problem (Peyré and Cuturi, 2019). To make $\mathcal{W}(\hat{\mu}_N, \hat{\nu}_M)$ practical to compute, one remedy is to smooth the optimal transport plan between $\hat{\mu}_N$ and $\hat{\nu}_M$ with an entropic regularization term, resulting in the Sinkhorn distance that still requires to be estimated with an iterative procedure, whose convergence is sensitive to the entropic regularization coefficient (Cuturi, 2013; Genevay et al., 2016; 2018; Xie et al., 2020). When the entropic regularization coefficient goes to infinity, we recover maximum mean discrepancy (MMD), which is considered as the metric for minimization, evaluated in a kernel space found by the adversarial mechanism in MMD-GAN (Li et al., 2015; 2017). By contrast, equipped with two navigators, the ACT can directly compute a forward point-to-distribution transport cost, denoted as $\mathcal{C}_{\boldsymbol{\phi},\boldsymbol{\theta}}(\boldsymbol{x} \to \hat{\nu}_M)$ in (8), and a backward one, denoted as $C_{\boldsymbol{\phi},\boldsymbol{\theta}}(\hat{\mu}_N \leftarrow \boldsymbol{y})$ in (10), which are then combined to define an unbiased sample estimator, as shown in (13), of the ACT divergence. Intuitively, the navigators play the role of "amortizing" the computation of the conditional transport plans between two empirical distributions, removing the need of using an iterative procedure to estimate the transport cost. From this amortization perspective, the navigators for ACT are analogous to the critic for the Wasserstein distance in its dual form.

**Lemma 5.** *Another unbiased sample estimator fully using the data in mini-batches* $\boldsymbol{x}_{1:N}$ *and* $\boldsymbol{y}_{1:M}$, *computing an amortized transport cost between two empirical distributions, can be expressed as*

$$\mathcal{L}_{\boldsymbol{\phi},\boldsymbol{\theta}}(\boldsymbol{x}_{1:N}, \boldsymbol{y}_{1:M}) = \sum_{i=1}^N \sum_{j=1}^M c(\boldsymbol{x}_i, \boldsymbol{y}_j) \left( \tfrac{1}{2N} \hat{\pi}_M(\boldsymbol{y}_j \,|\, \boldsymbol{x}_i, \boldsymbol{\phi}) + \tfrac{1}{2M} \hat{\pi}_N(\boldsymbol{x}_i \,|\, \boldsymbol{y}_j, \boldsymbol{\phi}) \right). \tag{14}$$

### 2.3 CRITIC BASED ADVERSARIAL FEATURE EXTRACTION

A naive definition of the transport cost between $\boldsymbol{x}$ and $\boldsymbol{y}$ is some distance between their raw feature vectors, such as $c(\boldsymbol{x}, \boldsymbol{y}) = \|\boldsymbol{x} - \boldsymbol{y}\|_2^2$, which, however, often poorly reflects the difference between high-dimensional data residing on low-dimensional manifolds. For this reason, with cosine dissimilarity (Salimans et al., 2018), we introduce a critic $\mathcal{T}_{\boldsymbol{\eta}}(\cdot)$, parameterized by $\boldsymbol{\eta}$, to help define an adversarial transport cost between two high-dimensional data points, expressed as

$$c_{\boldsymbol{\eta}}(\boldsymbol{x}, \boldsymbol{y}) = 1 - \cos(\mathcal{T}_{\boldsymbol{\eta}}(\boldsymbol{x}), \mathcal{T}_{\boldsymbol{\eta}}(\boldsymbol{y})), \quad \cos(\boldsymbol{h}_1, \boldsymbol{h}_2) \stackrel{\text{def.}}{=} \frac{|\boldsymbol{h}_1^T \boldsymbol{h}_2|}{\sqrt{\boldsymbol{h}_1^T \boldsymbol{h}_1} \sqrt{\boldsymbol{h}_2^T \boldsymbol{h}_2}}. \tag{15}$$

Intuitively, to minimize the distribution-to-distribution transport cost, the generator tries to mimic true data and both navigators try to optimize conditional path distributions. By contrast, the critic does the opposite by inflating the point-to-point transport cost.

In summary, given the training data set $\mathcal{X}$, to train the generator $G_{\boldsymbol{\theta}}$, forward navigator $\pi_{\boldsymbol{\phi}}(\boldsymbol{y} \mid \boldsymbol{x})$, backward navigator $\pi_{\boldsymbol{\phi}}(\boldsymbol{x} \mid \boldsymbol{y})$, and critic $\mathcal{T}_{\boldsymbol{\eta}}$, we propose to solve a mini-max problem as

$$\min_{\boldsymbol{\phi}, \boldsymbol{\theta}} \max_{\boldsymbol{\eta}} \mathbb{E}_{\boldsymbol{x}_{1:N} \subseteq \mathcal{X}, \ \boldsymbol{\epsilon}_{1:M} \stackrel{iid}{\sim} p(\boldsymbol{\epsilon})} [\mathcal{L}_{\boldsymbol{\phi}, \boldsymbol{\theta}, \boldsymbol{\eta}}(\boldsymbol{x}_{1:N}, \{G_{\boldsymbol{\theta}}(\boldsymbol{\epsilon}_j)\}_{j=1}^M)], \tag{16}$$

where $\mathcal{L}_{\boldsymbol{\phi}, \boldsymbol{\theta}, \boldsymbol{\eta}}$ is defined the same as in (13) or (14), except that we replace $c(\boldsymbol{x}_i, \boldsymbol{y}_j)$ in them with $c_{\boldsymbol{\eta}}(\boldsymbol{x}_i, \boldsymbol{y}_j)$ shown in (15) and draw $\boldsymbol{y}_{1:M}$ using reparameterization as in (6), which means $\boldsymbol{y}_{1:M} \stackrel{\text{def.}}{=} \{G_{\boldsymbol{\theta}}(\boldsymbol{\epsilon}_j)\}_{j=1}^M$. We train $\boldsymbol{\phi}$ and $\boldsymbol{\theta}$ with SGD using $\nabla_{\boldsymbol{\phi}, \boldsymbol{\theta}} \mathcal{L}_{\boldsymbol{\phi}, \boldsymbol{\theta}, \boldsymbol{\eta}}(\boldsymbol{x}_{1:N}, \{G_{\boldsymbol{\theta}}(\boldsymbol{\epsilon}_j)\}_{j=1}^M))$ and, if the critic is employed, train $\boldsymbol{\eta}$ with stochastic gradient ascent using $\nabla_{\boldsymbol{\eta}} \mathcal{L}_{\boldsymbol{\phi}, \boldsymbol{\theta}, \boldsymbol{\eta}}(\boldsymbol{x}_{1:N}, \{G_{\boldsymbol{\theta}}(\boldsymbol{\epsilon}_j)\}_{j=1}^M))$.

Note in existing critic-based GANs, how well the critics are optimized are directly related to how accurate and stable the gradients can be estimated. By contrast, regardless of how well the critic is optimized to inflate $c_{\boldsymbol{\eta}}(\boldsymbol{x}, \boldsymbol{y})$, Lemma 4 shows the sample estimate $\mathcal{L}_{\boldsymbol{\phi}, \boldsymbol{\theta}, \boldsymbol{\eta}}(\boldsymbol{x}_{1:N}, \{G_{\boldsymbol{\theta}}(\boldsymbol{\epsilon}_j)\}_{j=1}^M)$ of the ACT divergence and its gradients stay unbiased. Thus in ACT one can also train the critic parameter $\boldsymbol{\eta}$ using a different loss other than (16), such as the cross-entropy discriminator loss used in vanilla GANs to discriminate between $\boldsymbol{x}$ and $G_{\boldsymbol{\theta}}(\boldsymbol{\epsilon})$. This point will be verified in our ablation study.

## 3 EXPERIMENTAL RESULTS

**CT divergence for toy data:** As a proof of concept, we illustrate optimization under the CT divergence in 1D, with $x, y, \phi, \theta \in \mathbb{R}$. We consider a univariate normal distribution based example:

$$p(x) = \mathcal{N}(0, 1), \ \ p_\theta(y) = \mathcal{N}(0, e^\theta), \ \ c(x, y) = (x - y)^2, \ \ d(\mathcal{T}_\phi(x), \mathcal{T}_\phi(y)) = \frac{(x-y)^2}{2e^\phi}. \tag{17}$$

Thus $\theta = 0$ is the optimal solution that makes $\nu = \mu$. Denote $\sigma(a) = 1/(1 + e^{-a})$ as the sigmoid function, we have analytic forms of the Wasserstein distance as $\mathcal{W}_2(\mu, \nu)^2 = (1 - e^{\frac{\theta}{2}})^2$, forward and backward navigators as $\pi_\phi(y \mid x) = \mathcal{N}(\sigma(\theta - \phi)x, \sigma(\theta - \phi)e^\phi)$ and $\pi_\phi(x \mid y) = \mathcal{N}(\sigma(-\phi)y, \sigma(\phi))$, and forward and backward CT costs as $\mathcal{C}_{\phi, \theta}(\mu \to \nu) = \sigma(\phi - \theta)(e^\theta + \sigma(\phi - \theta))$ and $\mathcal{C}_{\phi, \theta}(\mu \leftarrow \nu) = \sigma(\phi)(1 + \sigma(\phi)e^\theta)$ (see Appendix B.1 for more details). Thus when applying gradient descent to minimize the CT divergence $\mathcal{C}_{\phi, \theta}(\mu, \nu)$, we expect the generator parameter $\theta \to 0$ as long as the learning rate of the navigator parameter $\phi$ is appropriately controlled to prevent $e^\phi \to 0$ from happening too soon. This is confirmed by Fig. 1, which shows that long before $e^\phi$ approaches zero, $\theta$ has already converged close to zero. This suggests that the navigator parameter $\phi$ mainly plays the role in assisting the learning of $\theta$. It is also interesting to observe that the CT divergences keep descending towards zero even when $\mathcal{W}_2(\mu, \nu)^2$ has already reached close to zero. As $\theta$ and $\phi$ converge towards their optimal solutions under the CT divergence, we can observe that $\mathcal{C}_{\phi, \theta}(\mu \to \nu)$ and $\mathcal{C}_{\phi, \theta}(\mu \leftarrow \nu)$ are getting closer. Moreover, $\mathcal{C}_{\phi, \theta}(\mu, \nu)$ initially stays above $\mathcal{W}_2(\mu, \nu)^2 = (1 - e^{\frac{\theta}{2}})^2$ but eventually becomes very close to $\mathcal{W}_2(\mu, \nu)^2$, which agrees what Lemma 2 suggests. The second and third subplots describe the descent trace on the gradient of the CT cost with respect to (*w.r.t.*) $\theta$ and $\phi$, respectively, while the fourth to six subplots show the forward, backward, and bi-directional CT costs, respectively, against $\theta$ when $e^\phi$ is optimized close to its optimum (see Fig. 8 for analogous plots for additional values of $e^\phi$). It is interesting to notice that the forward cost is minimized at $e^\theta > 1$, which implies mode covering, and the backward cost is minimized at $e^\theta \to 0$, which implies mode seeking, while the bi-directional cost is minimized at around the optimal solution $e^\theta = 1$; the forward CT cost

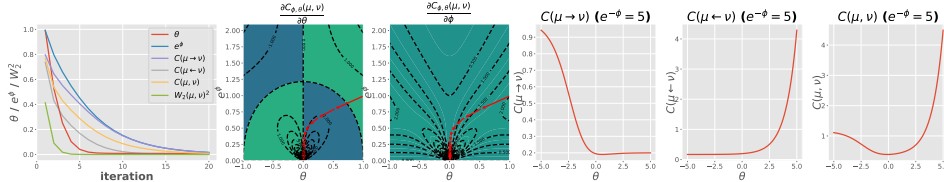

Figure 1: Illustration of minimizing the CT divergence $\mathcal{C}_{\phi,\theta}(\mu,\nu)$ between $\mathcal{N}(0,1)$ and $\mathcal{N}(0,e^{\theta})$. *Left*: Evolution of the CT divergence, its parameters and forward and backward costs, and corresponding Wasserstein distance; *Middle*: Gradients of the CT *w.r.t.* $\theta$ or $\phi$. The 2D trace of $(\theta, e^{\phi})$ is marked with red arrows. *Right*: The forward, backward, and CT values against $\theta$ when $e^{\phi}$ is optimized to a small value, which show combining forward and backward balances mode covering and seeking, making it easier for $\theta$ to move towards its optimum.

exhibits a flattened curve on the right hand side of its minimum, adding the backward CT cost to which not only moves that minimum left, making it closer to $\theta = 0$, but also raises the whole curve on the right hand side, making the optimum of $\theta$ become easier to reach via gradient descent.

We further consider a 1D example to illustrate the properties of the conditional transport distributions of the navigators, and analyze the risk for them to degenerate to point mass distributions when optimized under the CT or ACT divergence. We defer the details to Appendix B.2 and Fig. 9.

**ACT for 1D toy data:** We move on to model the empirical samples from a true data distribution, for which it is natural to apply the ACT divergence. To parameterize ACT, we apply a deep neural network to generator $G_{\theta}$ and another one to $\mathcal{T}_{\phi}$ that is shared by both navigators. We consider the squared Euclidean (*i.e.* $\mathcal{L}_2^2$) distance to define both cost $c(\boldsymbol{x}, \boldsymbol{y})$ and distance $d(\boldsymbol{h}_1, \boldsymbol{h}_2)$.

We first consider a 1D example, where $\mathcal{X}$ consists of $|\mathcal{X}| = 5,000$ samples $x_i \in \mathbb{R}$ of a bimodal Gaussian mixture $p_X(x) = \frac{1}{4}\mathcal{N}(x; -5, 1) + \frac{3}{4}\mathcal{N}(x; 2, 1)$. We illustrate in Fig. 2 the training with unbiased sample gradients $\nabla_{\phi,\theta}\mathcal{L}_{\phi,\theta}(\mathcal{X}, y_{1:M})$ of the ACT divergence shown in (14), where $y_j = G_{\theta}(\boldsymbol{\epsilon}_j)$. The top panel shows the ACT divergence, its backward and forward costs, and Wasserstein distance between the empirical probability measures $\hat{\mu}_N$ and $\hat{\nu}_M$ defined as in (5) and (6). We set $M = N$ and hence $\mathcal{W}_2(\hat{\mu}_X, \hat{\nu}_Y)^2$ can be exactly computed by sorting the 1D elements of $x_{1:N}$ and $y_{1:N}$ (Peyré and Cuturi, 2019). We first consider $N = |\mathcal{X}| = 5000$. Fig. 2 (Top) shows that the ACT divergence converges close to $\mathcal{W}_2(\hat{\mu}_X, \hat{\nu}_Y)^2$ and the forward and backward costs move closer to each other and can sometime go below

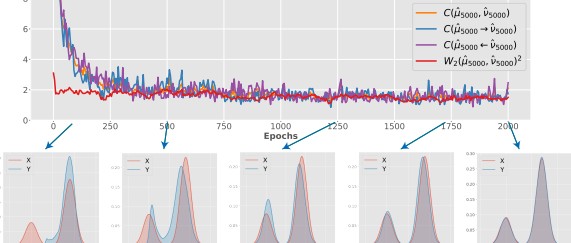

Figure 2: Illustration of how minimizing the ACT divergence between the empirical distribution of a generator and that of a bimodal Gaussian mixture, whose 5000 random samples are given, helps optimize the generator distribution towards the true one. *Top*: Plots of the ACT divergence $\mathcal{C}(\hat{\mu}_N, \hat{\nu}_M)$, forward ACT cost $\mathcal{C}(\hat{\mu}_N \rightarrow \hat{\nu}_M)$, backward ACT cost $\mathcal{C}(\hat{\mu}_N \leftarrow \hat{\nu}_M)$, and Wasserstein distance $\mathcal{W}_2(\hat{\mu}_N, \hat{\nu}_M)^2$, where $N = M = 5000$. *Bottom*: The PDF of the true data distribution $\mu(\mathrm{d}x) = p_X(x)\mathrm{d}x$ (red) and the generator distribution $\nu(\mathrm{d}y) = p_{\theta}(y)\mathrm{d}y$ (blue, visualized via kernel density estimation) at different training iterations.

$\mathcal{W}_2(\hat{\mu}_X, \hat{\nu}_Y)^2$. Fig. 2 (Bottom) shows that minimizing the ACT divergence successfully drives the generator distribution towards true data density: From the left to right, we can observe that initially the generator is focused on fitting a single mode; at around the $500^{th}$ iteration, as the forward and backward navigators are getting better, they start to help the generator locate the missing mode and we can observe a blue density mode starts to form over there; as the generator and both navigators are getting optimized, we can observe that the generator clearly captures both modes and the fitting is getting improved further; finally the generator well approximates the data density. Under the guidance of the ACT divergence, the generator and navigators are helping each other: An optimized generator helps the two navigators to train and realize the missing mode, and the optimized navigators help the generator locate under-fitted regions and hence better fit the true data density. Given the same $\mathcal{X}$, below we further consider setting $N = 20, 200$, or $5000$ to train the generator, using either the Wasserstein distance $\mathcal{W}_2(\hat{\mu}_N, \hat{\nu}_N)^2$ or ACT divergence $\mathcal{L}_{\phi,\theta}(x_{1:N}, y_{1:N})$ as the loss function.

As shown in the right column of Fig. 3, when the mini-batch size $N$ is as large as 5000, both Wasserstein and ACT lead to a well-trained generator. However, as shown in the left and middle columns, when $N$ is getting much smaller, we can observe that the generator trained with Wasserstein

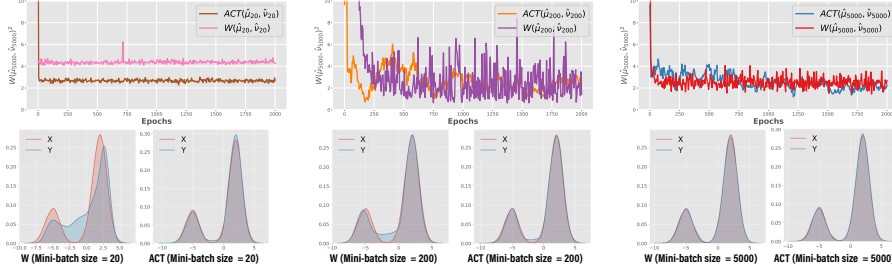

Figure 3: *Top*: Plot of the sample Wasserstein distance $W_2(\hat{\mu}_{5000}, \hat{\nu}_{5000})^2$ against the number of training epochs, where the generator is trained with either $W_2(\hat{\mu}_N, \hat{\nu}_N)^2$ or the ACT divergence between $\hat{\mu}_N$ and $\hat{\nu}_N$, with the mini-batch size set as $N = 20$ (left), $N = 200$ (middle), or $N = 5000$ (right); one epoch consists of $5000/N$ SGD iterations. *Bottom*: The fitting results of different configurations, where the KDE curves of the data distribution and the generative one are marked in red and blue, respectively.

clearly underperforms that trained with ACT, especially when the mini-batch size becomes as small as $N = 20$. While the Wasserstein distance $\mathcal{W}(\mu, \nu)$ in theory can well guide the training of a generative model, the sample Wasserstein distance $\mathcal{W}(\hat{\mu}_N, \hat{\nu}_N)$, whose optimal transport plan is locally re-computed for each mini-batch, could be sensitive to the mini-batch size $N$, which also explains why in practice the sample Wasserstein-based generative models are difficult to train and desire a large mini-batch size (Salimans et al., 2018). On the contrary, ACT amortizes its conditional transport plans through its navigators, whose parameter $\phi$ is globally updated across mini-batches, leading to a well-trained generator whose performance has low sensitivity to the mini-batch size.

**ACT for 2D toy data:** We further conduct experiments on four representative 2D datasets: 8-Gaussian mixture, Swiss Roll, Half Moons, and 25-Gaussian mixture, whose results are shown in Figs. 10-13 of Appendix B.3. We apply the vanilla GAN (Goodfellow et al., 2014) and Wasserstein GAN with gradient penalty (WGAN-GP) (Gulrajani et al., 2017) as two representatives of mini-max DGMs that require solving a mini-max loss to train the generator. We then apply the generators trained under the sliced Wasserstein distance (SWD) (Deshpande et al., 2018) and ACT divergence as two representatives of mini-max-free DGMs. Compared to mini-max DGMs, which require an adversarially learned critic in order to train the generator, one clear advantage of mini-max-free DGMs is that the generator is stable to train without the need of an adversarial game. On each 2D data, we train these DGMs as one would normally do during the first $15k$ iterations. We then only train the generator and freeze all the other learnable model parameters, which means we freeze the discriminator in GAN, critic in WGAN, and the navigator parameter $\phi$ of the ACT divergence, for another $15k$ iterations. Fig. 10 illustrates this training process on the 8-Gaussian mixture dataset, where for both mini-max DGMs, the mode collapse issue deteriorates after the first $15k$ iterations, while the training for SWD remains stable and that for ACT continues to improve. Compared to SWD, our method covers all 8 data density modes and moves the generator much closer to the true data density. On the other three datasets, the ACT divergence based DGM also exhibits good training consistency, high stability, and and close-to-optimal data generation, as shown in Appendix B.3.

**Resistance to mode collapse:** We use a 8-Gaussian mixture to empirically evaluate how well a DGM resists mode collapse. Unlike the data in Fig. 10, where 8 modes are equally weighted, here the mode at the left lower corner is set to have weight $\rho$ while the other modes are set to have the same weight of $\frac{1-\rho}{7}$. We set $\mathcal{X}$ with 5000 samples and the mini-batch size as $N = 100$. When $\rho$ is lowered to 0.05, its corresponding mode

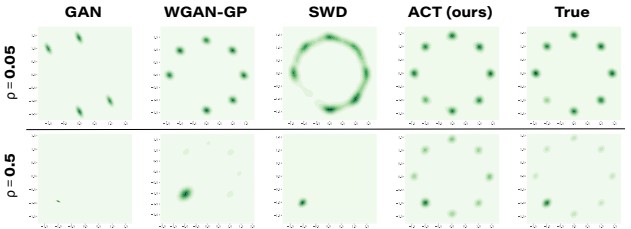

Figure 4: Comparison of the generation quality on 8-Gaussian mixture data: one of the 8 modes has weight $\rho$ and the rest modes have equal weight as $\frac{1-\rho}{7}$.

is shown to be missed by GAN, WGAN, and SWD-based DGM, while well kept by the ACT-based DGM. As an explanation, GANs are known to be susceptible to mode collapse; WGAN and SWD-based DGMs are sensitive to the mini-batch size, as when $\rho$ equals to a small value, the samples from this mode will appear in the mini-batches less frequently than those from any other mode, amplifying their missing mode problem. Similarly, when $\rho$ is increased to 0.5, the other modes are likely to be

missed by the baseline DGMs, while the ACT-based DGM does not miss any modes. The resistance of ACT to mode dropping can be attributed to the amortized computation of its conditional transport plans provided by the navigators, whose parameter is optimized with SGD over mini-batches and, as indicated by Fig. 3, is robust to estimate across a wide rage of mini-batch sizes.

**Forward and backward analysis:** To empirically analyze the roles played by the forward and backward ACTs in training a DGM, we modify (12) to define $\text{ACT}_\gamma$, where $\gamma \in [0, 1]$ is the interpolation weight from the forward ACT cost to the backward one, which means $\text{ACT}_\gamma$ reduces to the forward ACT when $\gamma = 1$, to backward ACT when $\gamma = 0$, and to the ACT in (12) when $\gamma = 0.5$. Fig. 5 shows the fitting results of $\text{ACT}_\gamma$ on the same 1D bi-modal Gaussian mixture used in Fig. 2 and 2D 8-Gaussian mixture used in Fig. 10; the other experimental

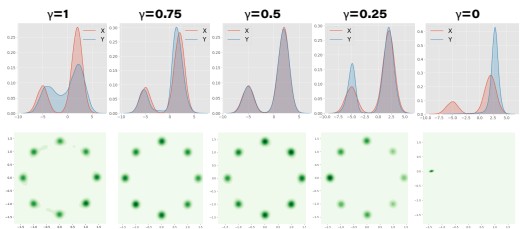

Figure 5: Fitting 1D bi-modal Gaussian (*top*) and 2D 8-Gaussian mixture (*bottom*) by interpolating between the forward ACT ($\gamma = 1$) and backward ACT ($\gamma = 0$).

settings are kept the same. Comparing the results of different $\gamma$ in Fig. 5 suggests that minimizing the forward transport cost only encourages the generator to exhibit mode covering behaviors, while minimizing the backward transport cost only encourages mode seeking/dropping behaviors; by contrast, combining both costs provides a user-controllable balance between mode covering and seeking, leading to satisfactory fitting performance, as shown in columns 2 to 4. Note that for a fair comparison, we stop the fitting at the same iteration; in practice, we find if training with more iterations, both $\gamma = 0.75$ and $\gamma = 0.25$ can achieve comparable results as $\gamma = 0.5$. Allowing the mode covering and seeking behaviors to be controlled by tuning $\gamma$ is an attractive property of $\text{ACT}_\gamma$. We leave the theoretical analysis of the mode covering/seeking behaviors of $\text{ACT}_\gamma$ for future study.

**ACT for natural images:** We conduct a variety of experiments on natural images to evaluate the performance and reveal the properties of DGMs optimized under the ACT divergence. We consider three widely-used image datasets, including CIFAR-10 (Krizhevsky et al., 2009), CelebA (Liu et al., 2015), and LSUN-bedroom (Yu et al., 2015), and compare the results of DGMs optimized with the ACT divergence against DGMs trained with the vanilla GAN and its various generalizations.

Note different from previous experiments on toy data, where the transport cost $c(\boldsymbol{x}, \boldsymbol{y})$ can be defined by directly comparing $\boldsymbol{x}$ and $\boldsymbol{y}$, for natural images, whose differences in raw pixel values are often not that meaningful, we need to compare $\mathcal{T}_{\boldsymbol{\eta}}(\boldsymbol{x})$ and $\mathcal{T}_{\boldsymbol{\eta}}(\boldsymbol{y})$, where $\mathcal{T}_{\boldsymbol{\eta}}(\cdot)$ is a critic that plays the role of adversarial feature extraction, as discussed in Section 2.3. In particular, we use (15) to define the transport cost as $c_{\boldsymbol{\eta}}(\boldsymbol{x}, \boldsymbol{y}) = 1 - \cos(\mathcal{T}_{\boldsymbol{\eta}}(\boldsymbol{x}), \mathcal{T}_{\boldsymbol{\eta}}(\boldsymbol{y}))$. To parameterize the navigators, we also set $d(\mathcal{T}_{\boldsymbol{\phi}}(\boldsymbol{x}), \mathcal{T}_{\boldsymbol{\phi}}(\boldsymbol{y})) = 1 - \cos(\mathcal{T}_{\boldsymbol{\phi}}(\boldsymbol{x}), \mathcal{T}_{\boldsymbol{\phi}}(\boldsymbol{y}))$. We test with the architecture suggested in Radford et al. (2015) as a standard CNN backbone and also apply the architecture in Miyato et al. (2018) as the ResNet (He et al., 2016) backbone. Specifically, we use the same architecture for the generator, and slightly modify the output dimension of the discriminator architecture as $2048$ for both $\mathcal{T}_{\boldsymbol{\eta}}$ of the critic and $\mathcal{T}_{\boldsymbol{\phi}}$ used by the two navigators. We train this model with (16) and (14), and to keep close to the corresponding backbone's original experiment setting, we set $N = M = 64$ for all experiments.

We summarize in Table 1 the Fréchet inception distance (FID) of Heusel et al. (2017) on all datasets and Inception Score (IS) of Salimans et al. (2016) on CIFAR-10. Both FID and IS are calculated using a pre-trained inception model (Szegedy et al., 2016). Lower FID and higher IS scores indicate better image quality. We observe that ACT-DCGAN and ACT-SNGAN, which are DCGAN and SNGAN backbones optimized with the ACT divergence, convincingly outperform DCGAN and SNGAN, respectively, suggesting that ACT is compatible with standard GANs and WGANs and generally helps improve generation quality. Moreover, ranked among the Top 3 on all datasets, ACT-SNGAN performs on par with the best benchmarks on CIFAR-10 and CelebA, while slightly worse on LSUN. The qualitative results shown in Fig. 6 are consistent with quantitative results in Table 1. To additionally show how ACT works for more complex generation tasks, we show in Fig. 7 example higher-resolution images generated by ACT-SNGAN on LSUN bedroom and CelebA-HQ.

Note ACT brings consistent improvement to DCGAN and SNGAN by neither improving their network architectures nor gradient regularization. Thus it has great potential to work in conjunction with other state-of-the-art architectures or methods, such as BigGAN (Brock et al., 2018), self-attention GANs (Zhang et al., 2019), BigBiGAN (Donahue and Simonyan, 2019), progressive training (Karras et al.,

Table 1: Comparison of generative models on CIFAR-10, CelebA and LSUN. (Top-3 are in bold)

| Method | Fréchet Inception Distance (FID, lower is better) | | | Inception Score (higher is better) |
|---|---|---|---|---|
| | CIFAR-10 | CelebA | LSUN-bedroom | CIFAR-10 |
| WGAN (Arjovsky et al., 2017) | 51.3± 1.5 | 37.1±1.9 | 73.3±2.5 | 6.9±0.1 |
| WGAN-GP (Gulrajani et al., 2017) | 19.0±0.8 | 18.0±0.7 | 26.9±1.1 | 7.9±0.1 |
| MMD-GAN (Li et al., 2017) | 73.9±0.1 | - | - | 6.2±0.1 |
| Cramér-GAN (Bellemare et al., 2017) | 40.3±0.2 | 31.3±0.2 | 54.2±0.4 | 6.4±0.1 |
| CTGAN (Wei et al., 2018) | **17.6±0.7** | 15.8±0.6 | **19.5±1.2** | 5.1±0.1 |
| OT-GAN (Salimans et al., 2018) | 32.5±0.6 | 19.4±3.0 | 70.5±5.3 | **8.5±0.1** |
| SWG (Deshpande et al., 2018) | 33.7±1.5 | 21.9±2.0 | 67.9±2.7 | - |
| Max-SWG (Deshpande et al., 2019) | 23.6±0.5 | **10.1±0.6** | 40.1±4.5 | - |
| SWGAN (Wu et al., 2019) | **17.0±1.0** | **13.2±0.7** | **14.9±1.0** | - |
| DCGAN (Radford et al., 2015) | 30.2±0.9 | 52.5±2.2 | 61.7±2.9 | 6.2±0.1 |
| DCGAN backbone + ACT divergence (ACT-DCGAN) | 24.8±1.0 | 29.2±2.0 | 37.4±2.5 | 7.5±0.1 |
| SNGAN (Miyato et al., 2018) | 21.5±1.3 | 21.7±1.5 | 31.1±2.1 | **8.2±0.1** |
| SNGAN backbone + ACT divergence (ACT-SNGAN) | **18.0±0.7** | **11.3±1.0** | **24.2±1.8** | **8.7±0.1** |

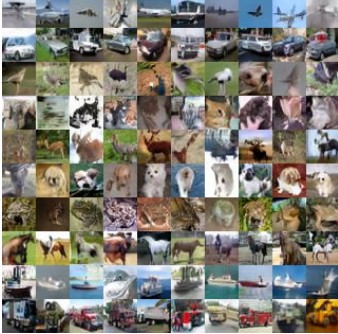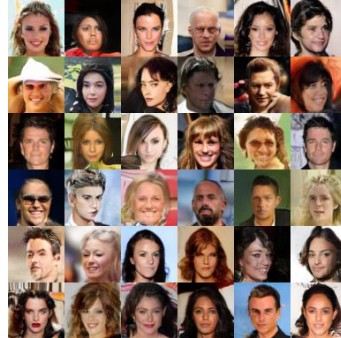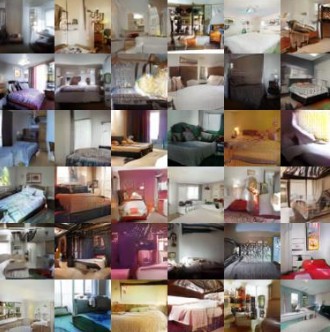

Figure 6: Generated samples of the deep generative model that adopts the backbone of SNGAN but is optimized with the ACT divergence on CIFAR-10, CelebA, and LSUN-Bedroom. See Appendix B for more results.

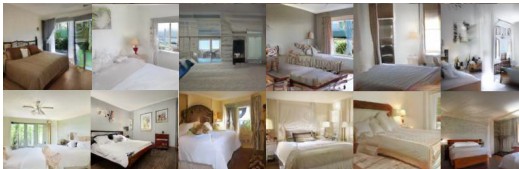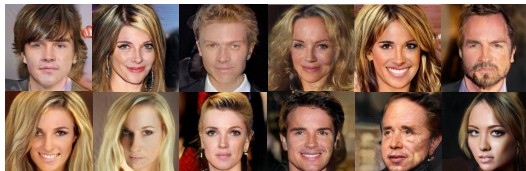

Figure 7: Analogous plot to Fig. 6 for *Left*: LSUN-Bedroom (128x128) and *Right*: CelebA-HQ (256x256).

2018), self-supervised learning (Chen et al., 2019), and data augmentation (Karras et al., 2020; Zhao et al., 2020a;b), which we leave for future study. As the paper is primarily focused on constructing and validating a new divergence measure, we have focused on demonstrating its efficacy on toy data and benchmark image data with moderate resolutions. We have focused on adapting DCGAN and SNGAN under ACT, and we leave to future work using the ACT to optimize a big DGM, such as BigGAN, that is often trained for high-resolution images with a substantially bigger network and larger mini-batch size and hence requires intensive computation that is not easy to afford.

## 4 CONCLUSION

We propose conditional transport (CT) as a new divergence to measure the difference between two probability distributions, via the use of both forward-path and backward-path point-to-point conditional distributions. To apply CT to two distributions that have unknown density functions but are easy to sample from, we introduce the asymptotic CT (ACT) divergence whose estimation only requires access to empirical samples. ACT amortizes the computation of its conditional transport plans via its navigators, removing the need of a separate iterative procedure for each mini-batch, and provides unbiased mini-batch based sample gradients that are simple to compute. Its minimization, achieved with the collaboration between the generator, forward navigator, and backward navigator, is shown to be robust to the mini-batch size. In addition, empirical analysis suggests the combination weight of the forward and backward CT costs can be adjusted to encourage either model covering or seeking behaviors. We further show that a critic can be integrated into the point-to-point transport cost of ACT to adversarially extract features from high-dimensional data without biasing the sample gradients. We apply ACT to train both a vanilla GAN and a Wasserstein GAN. Consistent improvement is observed in our experiments, which shows the potential of the ACT divergence in more broader settings where quantifying the difference between distributions plays an essential role.

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

## A  PROOFS

*Proof of Lemma 1.* If $\boldsymbol{y} \sim p_{\boldsymbol{\theta}}(\boldsymbol{y})$ is equal to $\boldsymbol{x} \sim p_X(\boldsymbol{x})$ in distribution, then $p_X(\boldsymbol{x}) = p_{\boldsymbol{\theta}}(\boldsymbol{x})$ for any $\boldsymbol{x} \in \mathbb{R}^V$. For (2) we have $\int p_X(\boldsymbol{x})\pi_{\boldsymbol{\phi}}(\boldsymbol{y} \,|\, \boldsymbol{x})\mathrm{d}\boldsymbol{y} = p_X(\boldsymbol{x}) \int \pi_{\boldsymbol{\phi}}(\boldsymbol{y} \,|\, \boldsymbol{x})\mathrm{d}\boldsymbol{y} = p_X(\boldsymbol{x})$ and

$$\int p_X(\boldsymbol{x})\pi_{\boldsymbol{\phi}}(\boldsymbol{y} \,|\, \boldsymbol{x})\mathrm{d}\boldsymbol{x} = \int p_X(\boldsymbol{x})\frac{e^{-d(\mathcal{T}_{\boldsymbol{\phi}}(\boldsymbol{x}),\mathcal{T}_{\boldsymbol{\phi}}(\boldsymbol{y}))}p_{\boldsymbol{\theta}}(\boldsymbol{y})}{\int e^{-d(\mathcal{T}_{\boldsymbol{\phi}}(\boldsymbol{x}),\mathcal{T}_{\boldsymbol{\phi}}(\boldsymbol{y}))}p_{\boldsymbol{\theta}}(\boldsymbol{y})\mathrm{d}\boldsymbol{y}}\mathrm{d}\boldsymbol{x}$$

$$= p_{\boldsymbol{\theta}}(\boldsymbol{y}) \int \frac{e^{-d(\mathcal{T}_{\boldsymbol{\phi}}(\boldsymbol{x}),\mathcal{T}_{\boldsymbol{\phi}}(\boldsymbol{y}))}p_X(\boldsymbol{x})}{\int e^{-d(\mathcal{T}_{\boldsymbol{\phi}}(\boldsymbol{x}),\mathcal{T}_{\boldsymbol{\phi}}(\boldsymbol{y}))}p_{\boldsymbol{\theta}}(\boldsymbol{y})\mathrm{d}\boldsymbol{y}}\mathrm{d}\boldsymbol{x}.$$

If $e^{-d(\mathcal{T}_{\boldsymbol{\phi}}(\boldsymbol{x}),\mathcal{T}_{\boldsymbol{\phi}}(\boldsymbol{y}))} = \mathbf{1}(\boldsymbol{x} = \boldsymbol{y})$, then we further have

$$\int \frac{e^{-d(\mathcal{T}_{\boldsymbol{\phi}}(\boldsymbol{x}),\mathcal{T}_{\boldsymbol{\phi}}(\boldsymbol{y}))}p_X(\boldsymbol{x})}{\int e^{-d(\mathcal{T}_{\boldsymbol{\phi}}(\boldsymbol{x}),\mathcal{T}_{\boldsymbol{\phi}}(\boldsymbol{y}))}p_{\boldsymbol{\theta}}(\boldsymbol{y})\mathrm{d}\boldsymbol{y}}\mathrm{d}\boldsymbol{x} = \int \frac{\mathbf{1}(\boldsymbol{x} = \boldsymbol{y})p_X(\boldsymbol{x})}{\int \mathbf{1}(\boldsymbol{x} = \boldsymbol{y})p_{\boldsymbol{\theta}}(\boldsymbol{y})\mathrm{d}\boldsymbol{y}}\mathrm{d}\boldsymbol{x} = \int \mathbf{1}(\boldsymbol{x} = \boldsymbol{y})\frac{p_X(\boldsymbol{x})}{p_{\boldsymbol{\theta}}(\boldsymbol{x})}\mathrm{d}\boldsymbol{x} = 1$$

and hence it is true that

$$\int p_X(\boldsymbol{x})\pi_{\boldsymbol{\phi}}(\boldsymbol{y} \,|\, \boldsymbol{x})\mathrm{d}\boldsymbol{x} = p_{\boldsymbol{\theta}}(\boldsymbol{y}).$$

Similarly, for (3) we have $\int p_{\boldsymbol{\theta}}(\boldsymbol{y})\pi_{\boldsymbol{\phi}}(\boldsymbol{x} \,|\, \boldsymbol{y})\mathrm{d}\boldsymbol{x} = p_{\boldsymbol{\theta}}(\boldsymbol{y})$ and can prove $\int p_{\boldsymbol{\theta}}(\boldsymbol{y})\pi_{\boldsymbol{\phi}}(\boldsymbol{x} \,|\, \boldsymbol{y})\mathrm{d}\boldsymbol{y} = p_X(\boldsymbol{x})$ given these two conditions. $\qquad\square$

*Proof of Lemma 2.* Since $c(\boldsymbol{x}, \boldsymbol{y}) \geq 0$ by definition, we have $\mathcal{C}_{\boldsymbol{\phi},\boldsymbol{\theta}}(\mu \rightarrow \nu) \geq 0$ and $\mathcal{C}_{\boldsymbol{\phi},\boldsymbol{\theta}}(\mu \leftarrow \nu) \geq 0$. When $\mu = \nu$, it is known that $\mathcal{W}(\mu, \nu) = 0$. If $\boldsymbol{y} \sim p_{\boldsymbol{\theta}}(\boldsymbol{y})$ is equal to $\boldsymbol{x} \sim p_X(\boldsymbol{x})$ in distribution, which means $p_X(\boldsymbol{x}) = p_{\boldsymbol{\theta}}(\boldsymbol{x})$ and $p_X(\boldsymbol{y}) = p_{\boldsymbol{\theta}}(\boldsymbol{y})$ for any $\boldsymbol{x}, \boldsymbol{y} \in \mathbb{R}^V$ and $\mu = \nu$, then we have

$$\mathcal{C}_{\boldsymbol{\phi},\boldsymbol{\theta}}(\mu \rightarrow \nu) = \int \int c(\boldsymbol{x}, \boldsymbol{y})p_X(\boldsymbol{x})\pi_{\boldsymbol{\phi}}(\boldsymbol{y} \,|\, \boldsymbol{x})\mathrm{d}\boldsymbol{x}\mathrm{d}\boldsymbol{y}$$

$$= \int \int c(\boldsymbol{x}, \boldsymbol{y})\frac{e^{-d(\mathcal{T}_{\boldsymbol{\phi}}(\boldsymbol{x}),\mathcal{T}_{\boldsymbol{\phi}}(\boldsymbol{y}))}p_X(\boldsymbol{x})p_{\boldsymbol{\theta}}(\boldsymbol{y})}{\int e^{-d(\mathcal{T}_{\boldsymbol{\phi}}(\boldsymbol{x}),\mathcal{T}_{\boldsymbol{\phi}}(\boldsymbol{y}))}p_{\boldsymbol{\theta}}(\boldsymbol{y})\mathrm{d}\boldsymbol{y}}\mathrm{d}\boldsymbol{x}\mathrm{d}\boldsymbol{y}$$

$$= \int \int c(\boldsymbol{y}, \boldsymbol{x})\frac{e^{-d(\mathcal{T}_{\boldsymbol{\phi}}(\boldsymbol{y}),\mathcal{T}_{\boldsymbol{\phi}}(\boldsymbol{x}))}p_X(\boldsymbol{y})p_{\boldsymbol{\theta}}(\boldsymbol{x})}{\int e^{-d(\mathcal{T}_{\boldsymbol{\phi}}(\boldsymbol{y}),\mathcal{T}_{\boldsymbol{\phi}}(\boldsymbol{x}))}p_{\boldsymbol{\theta}}(\boldsymbol{x})\mathrm{d}\boldsymbol{x}}\mathrm{d}\boldsymbol{x}\mathrm{d}\boldsymbol{y}$$

$$= \int \int c(\boldsymbol{y}, \boldsymbol{x})\frac{e^{-d(\mathcal{T}_{\boldsymbol{\phi}}(\boldsymbol{y}),\mathcal{T}_{\boldsymbol{\phi}}(\boldsymbol{x}))}p_{\boldsymbol{\theta}}(\boldsymbol{y})p_X(\boldsymbol{x})}{\int e^{-d(\mathcal{T}_{\boldsymbol{\phi}}(\boldsymbol{y}),\mathcal{T}_{\boldsymbol{\phi}}(\boldsymbol{x}))}p_X(\boldsymbol{x})\mathrm{d}\boldsymbol{x}}\mathrm{d}\boldsymbol{x}\mathrm{d}\boldsymbol{y}$$

$$= \int \int c(\boldsymbol{x}, \boldsymbol{y})p_{\boldsymbol{\theta}}(\boldsymbol{y})\frac{e^{-d(\mathcal{T}_{\boldsymbol{\phi}}(\boldsymbol{x}),\mathcal{T}_{\boldsymbol{\phi}}(\boldsymbol{y}))}p_X(\boldsymbol{x})}{\int e^{-d(\mathcal{T}_{\boldsymbol{\phi}}(\boldsymbol{x}),\mathcal{T}_{\boldsymbol{\phi}}(\boldsymbol{y}))}p_X(\boldsymbol{x})\mathrm{d}\boldsymbol{x}}\mathrm{d}\boldsymbol{x}\mathrm{d}\boldsymbol{y}$$

$$= \int \int c(\boldsymbol{x}, \boldsymbol{y})p_{\boldsymbol{\theta}}(\boldsymbol{y})\pi_{\boldsymbol{\phi}}(\boldsymbol{x} \,|\, \boldsymbol{y})\mathrm{d}\boldsymbol{x}\mathrm{d}\boldsymbol{y}$$

$$= \mathcal{C}_{\boldsymbol{\phi},\boldsymbol{\theta}}(\mu \leftarrow \nu)$$

and hence $C_{\boldsymbol{\phi},\boldsymbol{\theta}}(\mu, \nu) = \mathcal{C}_{\boldsymbol{\phi},\boldsymbol{\theta}}(\mu \rightarrow \nu) = \mathcal{C}_{\boldsymbol{\phi},\boldsymbol{\theta}}(\mu \leftarrow \nu) \geq 0 = W(\mu, \nu)$.

If $e^{-d(\mathcal{T}_{\boldsymbol{\phi}}(\boldsymbol{x}),\mathcal{T}_{\boldsymbol{\phi}}(\boldsymbol{y}))} = \mathbf{1}(\boldsymbol{x} = \boldsymbol{y})$, since $c(\boldsymbol{x}, \boldsymbol{x}) = 0$ by definition, we have

$$\mathcal{C}_{\boldsymbol{\phi},\boldsymbol{\theta}}(\mu \rightarrow \nu) = \int \int c(\boldsymbol{x}, \boldsymbol{y})\frac{\mathbf{1}(\boldsymbol{x} = \boldsymbol{y})p_X(\boldsymbol{x})p_{\boldsymbol{\theta}}(\boldsymbol{y})}{\int \mathbf{1}(\boldsymbol{x} = \boldsymbol{y})p_{\boldsymbol{\theta}}(\boldsymbol{y})\mathrm{d}\boldsymbol{y}}\mathrm{d}\boldsymbol{x}\mathrm{d}\boldsymbol{y}$$

$$= \int \int c(\boldsymbol{x}, \boldsymbol{y})\frac{\mathbf{1}(\boldsymbol{x} = \boldsymbol{y})p_X(\boldsymbol{x})p_{\boldsymbol{\theta}}(\boldsymbol{y})}{p_{\boldsymbol{\theta}}(\boldsymbol{x})}\mathrm{d}\boldsymbol{x}\mathrm{d}\boldsymbol{y}$$

$$= \int \int c(\boldsymbol{x}, \boldsymbol{y})\mathbf{1}(\boldsymbol{x} = \boldsymbol{y})p_{\boldsymbol{\theta}}(\boldsymbol{y})\mathrm{d}\boldsymbol{x}\mathrm{d}\boldsymbol{y}$$

$$= \int c(\boldsymbol{x}, \boldsymbol{x})p_{\boldsymbol{\theta}}(\boldsymbol{x})d\boldsymbol{x}$$

$$= 0.$$

$\qquad\square$

*Proof of Lemma 3.* According to the strong law of large numbers, when $M \to \infty$, $\hat{\nu}_M(A) = \frac{1}{M} \sum_{j=1}^{M} \mathbf{1}(\boldsymbol{y}_j \in A)$ converges almost surely to

$$\frac{1}{M} \sum_{j=1}^{M} \mathbb{E}_{\boldsymbol{y}_j \sim p_{\boldsymbol{\theta}}(\boldsymbol{y})} [\mathbf{1}(\boldsymbol{y}_j \in A)] = \int_A p_{\boldsymbol{\theta}}(\boldsymbol{y}) d\boldsymbol{y} = \nu(A)$$

and hence $C_{\boldsymbol{\phi},\boldsymbol{\theta}}(\mu \to \hat{\nu}_M)$ converges to $C_{\boldsymbol{\phi},\boldsymbol{\theta}}(\mu \to \nu)$. Therefore, $\mathbb{E}_{\boldsymbol{y}_{1:M} \overset{iid}{\sim} p_{\boldsymbol{\theta}}(\boldsymbol{y})}[C_{\boldsymbol{\phi},\boldsymbol{\theta}}(\mu \to \hat{\nu}_M)]$ converges to $C_{\boldsymbol{\phi},\boldsymbol{\theta}}(\mu \to \nu)$. Similarly, we can prove that as $N \to \infty$, $\mathbb{E}_{\boldsymbol{x}_{1:N} \overset{iid}{\sim} p_X(\boldsymbol{x})}[C_{\boldsymbol{\phi},\boldsymbol{\theta}}(\hat{\mu}_N \leftarrow \nu)]$ converges to $C_{\boldsymbol{\phi},\boldsymbol{\theta}}(\mu \leftarrow \nu)$. Therefore, $C_{\boldsymbol{\phi},\boldsymbol{\theta}}(\mu, \nu, N, M)$ defined in (12) converges to $\frac{1}{2} C_{\boldsymbol{\phi},\boldsymbol{\theta}}(\mu \to \nu) + \frac{1}{2} C_{\boldsymbol{\phi},\boldsymbol{\theta}}(\mu \leftarrow \nu) = C_{\boldsymbol{\phi},\boldsymbol{\theta}}(\mu, \nu)$ as $N, M \to \infty$. □

*Proof of Lemma 4.*

$$\begin{aligned} C_{\boldsymbol{\phi},\boldsymbol{\theta}}(\mu, \nu, N, M) &= \frac{1}{2} \mathbb{E}_{\boldsymbol{y}_{1:M} \overset{iid}{\sim} p_{\boldsymbol{\theta}}(\boldsymbol{y})} [C_{\boldsymbol{\phi},\boldsymbol{\theta}}(\mu \to \hat{\nu}_M)] + \frac{1}{2} \mathbb{E}_{\boldsymbol{x}_{1:N} \overset{iid}{\sim} p_X(\boldsymbol{x})} [C_{\boldsymbol{\phi},\boldsymbol{\theta}}(\hat{\mu}_N \leftarrow \nu)] \\ &= \frac{1}{2} \mathbb{E}_{\boldsymbol{x} \sim p_X(\boldsymbol{x}), \, \boldsymbol{y}_{1:M} \overset{iid}{\sim} p_{\boldsymbol{\theta}}(\boldsymbol{y})} [C_{\boldsymbol{\phi},\boldsymbol{\theta}}(\boldsymbol{x} \to \hat{\nu}_M)] + \frac{1}{2} \mathbb{E}_{\boldsymbol{x}_{1:N} \overset{iid}{\sim} p_X(\boldsymbol{x}), \, \boldsymbol{y} \sim p_{\boldsymbol{\theta}}(\boldsymbol{y})} [C_{\boldsymbol{\phi},\boldsymbol{\theta}}(\hat{\mu}_N \leftarrow \boldsymbol{y})] \\ &= \frac{1}{2} \mathbb{E}_{\boldsymbol{x} \sim \hat{p}_N(\boldsymbol{x})} \mathbb{E}_{\boldsymbol{x}_{1:M} \overset{iid}{\sim} p_X(\boldsymbol{x}), \, \boldsymbol{y}_{1:M} \overset{iid}{\sim} p_{\boldsymbol{\theta}}(\boldsymbol{y})} [C_{\boldsymbol{\phi},\boldsymbol{\theta}}(\boldsymbol{x} \to \hat{\nu}_M)] \\ &\quad + \frac{1}{2} \mathbb{E}_{\boldsymbol{y} \sim \hat{p}_M(\boldsymbol{y})} \mathbb{E}_{\boldsymbol{x}_{1:N} \overset{iid}{\sim} p_X(\boldsymbol{x}), \, \boldsymbol{y}_{1:M} \overset{iid}{\sim} p_{\boldsymbol{\theta}}(\boldsymbol{y})} [C_{\boldsymbol{\phi},\boldsymbol{\theta}}(\hat{\mu}_N \leftarrow \boldsymbol{y})] \\ &= \mathbb{E}_{\boldsymbol{x} \sim \hat{p}_N(\boldsymbol{x}), \, \boldsymbol{y} \sim \hat{p}_M(\boldsymbol{y})} \mathbb{E}_{\boldsymbol{x}_{1:N} \overset{iid}{\sim} p_X(\boldsymbol{x}), \, \boldsymbol{y}_{1:M} \overset{iid}{\sim} p_{\boldsymbol{\theta}}(\boldsymbol{y})} \left[ \frac{1}{2} C_{\boldsymbol{\phi},\boldsymbol{\theta}}(\boldsymbol{x} \to \hat{\nu}_M) + \frac{1}{2} C_{\boldsymbol{\phi},\boldsymbol{\theta}}(\hat{\mu}_N \leftarrow \boldsymbol{y}) \right]. \quad (18) \end{aligned}$$

Plugging (8) and (10) into the above equation concludes the proof.

□

*Proof of Lemma 5.* Solving the first expectation of (18), we have

$$\begin{aligned} &C_{\boldsymbol{\phi},\boldsymbol{\theta}}(\mu, \nu, N, M) \\ &= \mathbb{E}_{\boldsymbol{x}_{1:N} \overset{iid}{\sim} p_X(\boldsymbol{x}), \, \boldsymbol{y}_{1:M} \overset{iid}{\sim} p_{\boldsymbol{\theta}}(\boldsymbol{y})} \left[ \frac{1}{2N} \sum_{i=1}^{N} C_{\boldsymbol{\phi},\boldsymbol{\theta}}(\boldsymbol{x}_i \to \hat{\nu}_M) + \frac{1}{2M} \sum_{j=1}^{M} C_{\boldsymbol{\phi},\boldsymbol{\theta}}(\hat{\mu}_N \leftarrow \boldsymbol{y}_j) \right]. \end{aligned}$$

Plugging (8) and (10) into the above equation concludes the proof.

□

## B SUPPLEMENTARY EXPERIMENT RESULTS

### B.1 ADDITIONAL DETAILS FOR THE UNIVARIATE NORMAL TOY EXAMPLE SHOWN IN (17)

For the toy example specified in (17), exploiting the normal-normal conjugacy, we have an analytical conditional distribution for the forward navigator as

$$\begin{aligned} \pi_{\phi}(y \mid x) &\propto e^{-\frac{(x-y)^2}{2e^{\phi}}} \mathcal{N}(y; 0, e^{\theta}) \\ &\propto \mathcal{N}(x; y, e^{\phi}) \mathcal{N}(y; 0, e^{\theta}) \\ &= \mathcal{N}\left( \frac{e^{\theta}}{e^{\theta} + e^{\phi}} x, \frac{e^{\phi} e^{\theta}}{e^{\theta} + e^{\phi}} \right), \end{aligned}$$

and an analytical conditional distribution for the backward navigator as

$$\begin{aligned} \pi_{\phi}(x \mid y) &\propto e^{-\frac{(x-y)^2}{2e^{\phi}}} \mathcal{N}(x; 0, 1) \\ &\propto \mathcal{N}(y; x, e^{\phi}) \mathcal{N}(x; 0, 1) \\ &= \mathcal{N}\left( \frac{y}{1 + e^{\phi}}, \frac{e^{\phi}}{1 + e^{\phi}} \right). \end{aligned}$$

Plugging them into (2) and (3), respectively, and solving the expectations, we have

$$\mathcal{C}_{\phi,\theta}(\mu \to \nu) = \mathbb{E}_{x \sim \mathcal{N}(0,1)} \left[ \frac{e^{\phi}}{e^{\theta} + e^{\phi}} \left( e^{\theta} + \frac{e^{\phi}}{e^{\theta} + e^{\phi}} x^2 \right) \right]$$

$$= \frac{e^{\phi}}{e^{\theta} + e^{\phi}} \left( e^{\theta} + \frac{e^{\phi}}{e^{\theta} + e^{\phi}} \right),$$

$$\mathcal{C}_{\phi,\theta}(\mu \leftarrow \nu) = \mathbb{E}_{y \sim \mathcal{N}(0,e^{\theta})} \left[ \frac{e^{\phi}}{1 + e^{\phi}} \left( 1 + \frac{e^{\phi}}{1 + e^{\phi}} y^2 \right) \right]$$

$$= \frac{e^{\phi}}{1 + e^{\phi}} \left( 1 + \frac{e^{\phi}}{1 + e^{\phi}} e^{\theta} \right).$$

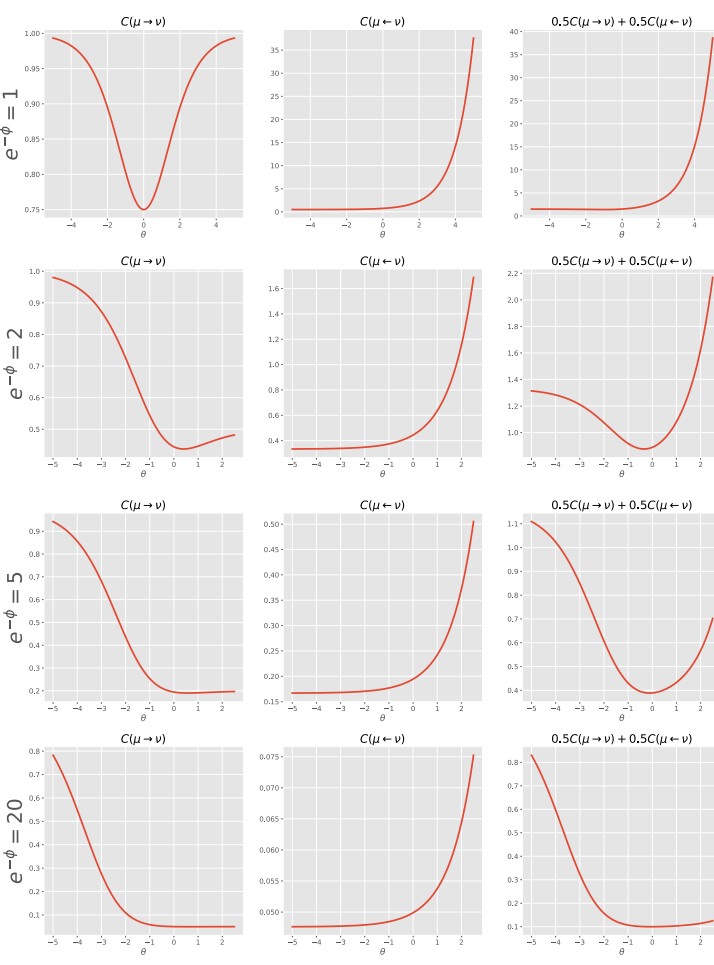

Figure 8: For the univariate normal based toy example specified in (17), we plot the forward, backward, and CT values against $\theta$ at four different values of $\phi$, which show combining forward and backward balances mode covering and seeking, making it easier for $\theta$ to move towards its optimum.

### B.2 ANALYSIS OF CONDITIONAL TRANSPORT PLANS ON 1D MIXTURE

We consider a 1D example to illustrate the properties of the conditional transport plans of the navigators, and analyze the risk for them to degenerate to point mass distributions when optimized under the CT or ACT divergence. We consider two representative scenarios that both seem to pose a high risk for the conditional transport distributions to degenerate. In the first scenario, we consider both the source and target distributions as a mixture of a point mass and a Gaussian distribution, where the location of the point mass and the center of the Gaussian are far away from each other; we obtain the analytic forms of the conditional transport plans under the CT divergence, and analyze the properties of the empirical conditional transport plans under the ACT divergence. In the second scenario, we consider both the source and target distributions as discrete distributions; we make the support of each distribution contain an outlier supporting point far away from all the other supporting points. In this scenario, we are essentially learning how to transport between two discrete sets.

#### B.2.1 CONDITIONAL TRANSPORT BETWEEN TWO POINT-MASS-AND-GAUSSIAN MIXTURE DISTRIBUTIONS

Below we consider the first scenario, where we assume

$$p_X(x) = \rho \delta_{-1} + (1 - \rho)N(-1000, 1)$$

and

$$p_Y(y) = \rho \delta_1 + (1 - \rho)N(1000, 1),$$

where $\rho \in [0, 1]$ is the probability for $x = -1$ and $y = 1$. We construct this specific example to check whether there is a danger that the forward CT distribution $\pi(y \mid x)$ will degenerate to a point mass distribution that concentrates its probability mass at $y = 1$. Setting $d(\mathcal{T}_\phi(x), \mathcal{T}_\phi(y)) = \frac{(x-y)^2}{2e^\phi}$, via the definition of the conditional transport distributions and the property of the normal distribution, we can show that

$$\pi(y \mid x, \phi) = \frac{\rho r(y, x)}{\rho r(y, x) + 1 - \rho} \delta_1 + \frac{1 - \rho}{\rho r(y, x) + 1 - \rho} \mathcal{N}(\mu, \sigma^2),$$

$$r(y, x) \overset{\text{def.}}{=} \frac{\mathcal{N}(y; \mu, \sigma^2)}{\mathcal{N}(y; 1000, 1)}, \quad \mu \overset{\text{def.}}{=} 1000 \frac{e^\phi}{1 + e^\phi} + x \frac{1}{1 + e^\phi}, \quad \sigma^2 \overset{\text{def.}}{=} \frac{e^\phi}{1 + e^\phi}. \tag{19}$$

Similarly, for the backward CT, we can show that

$$\pi(x \mid y, \phi) = \frac{\rho r'(y, x)}{\rho r'(y, x) + 1 - \rho} \delta_{-1} + \frac{1 - \rho}{\rho r'(y, x) + 1 - \rho} \mathcal{N}(\mu', \sigma^2),$$

$$r' \overset{\text{def.}}{=} \frac{\mathcal{N}(x; \mu', \sigma^2)}{\mathcal{N}(x; -1000, 1)}, \quad \mu' \overset{\text{def.}}{=} -1000 \frac{e^\phi}{1 + e^\phi} + y \frac{1}{1 + e^\phi}, \quad \sigma^2 \overset{\text{def.}}{=} \frac{e^\phi}{1 + e^\phi}. \tag{20}$$

As shown in the top panel of Fig. 9, it is clear that when $\phi \to \infty$, we have $\pi(y = 1 \mid x) = \rho$ and when $\phi \to -\infty$, we have $\pi(y = 1 \mid x) = 0$. Assuming $\rho = 0.01$ and $x = -1000$, when $\phi = 15$, we have $\pi_\phi(y = 1 \mid x = -1000) = 0.0157$ and $\pi_\phi(y = 1 \mid x = -1) = 0.0116$; when $\phi = -1.095945$, we have $\pi_\phi(y = 1 \mid x = -1000) = 0.0626$ and $\pi_\phi(y = 1 \mid x = -1) = 1$. These analyses suggest that as long the navigator parameter $\phi$ is chosen appropriately, the conditional transport distributions $\pi(x \mid y, \phi)$ and $\pi(y \mid x, \phi)$ will not degenerate to a point mass distribution.

Next we analyze the degeneration risk if using the ACT between $p_X(x) = \rho \delta_{-1} + (1 - \rho)N(-1000, 1)$ and $p_Y(y) = \rho \delta_1 + (1 - \rho)N(1000, 1)$, in which case we assume we don't know $p_X(x)$ and $p_Y(y)$ but can access random samples from them. In ACT, we would approximate $\pi(y \mid x)$ with

$$\hat{\pi}_M(y \mid x) \overset{\text{def.}}{=} \sum_{j=1}^{M} \frac{e^{-\frac{(x-y_j)^2}{2e^\phi}}}{\sum_{j'=1}^{M} e^{-\frac{(x-y_j)^2}{2e^\phi}}} \delta_{y_j}, \quad y_j \overset{iid}{\sim} p_Y(y). \tag{21}$$

Even if $\phi$ is set at a value such that the forward CT distribution $\pi_\phi(y = 1 \mid x) \to 1$, under ACT, as long as $1 \notin \{y_1, \ldots, y_M\}$, $x$ will surely not be transported to $y = 1$ and will instead be transported to a $y_j$ that is drawn from $\mathcal{N}(1000, 1)$; note $1 \notin \{y_1, \ldots, y_M\}$ happens with probability $\prod_{j=1}^{M} P(y_j \neq 1) = (1 - \rho)^M$, which is as large as 36.6% when $\rho = 0.01$ and $M = 100$. Therefore, mini-batch based optimization under the ACT divergence would further reduce the risk for the conditional transport plans to degenerate.

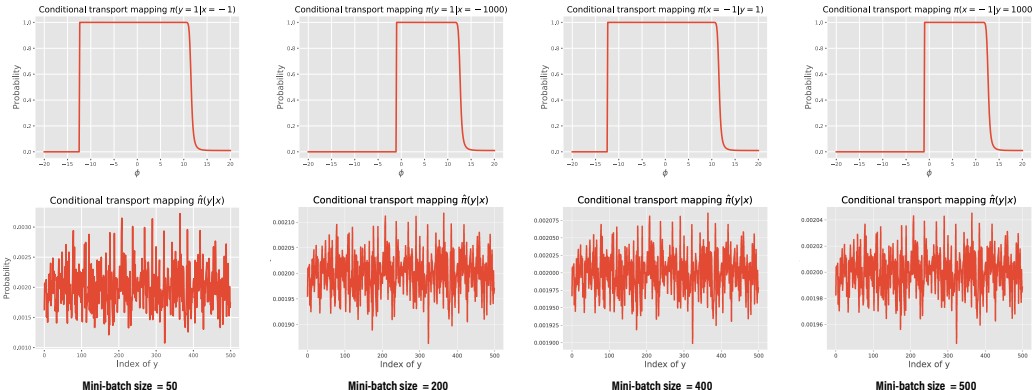

Figure 9: *Top*: The risk for the forward and backward conditional transport plans to degenerate *w.r.t.* the value of $\phi$ based on CT analysis. *Bottom*: The forward conditional transport plan between two discrete sets of $S = 500$ points; among the $S$ data points, one point is an outlider. The ACT navigators are optimized with SGD over mini-batches, whose elements are randomly sampled (with replacement) from their corresponding discrete sets and the mini-batch size varies from 50 to 500.

### B.2.2 ACT BETWEEN TWO DISCRETE DISTRIBUTIONS WITH OUTLIER SUPPORTS

Below we consider the second scenario, where we assume two discrete distributions supported on $S = 500$ data points as

$$p_X(x) = \frac{1}{S}\delta_{-1} + \frac{S-1}{S}\sum_{i=1}^{S-1}\delta_{x_i},$$

where $x_i \overset{iid}{\sim} \mathcal{N}(-1000, 1)$, and

$$p_Y(y) = \frac{1}{S}\delta_{1} + \frac{S-1}{S}\sum_{i=1}^{S-1}\delta_{y_i},$$

where $y_i \overset{iid}{\sim} \mathcal{N}(1000, 1)$. In this scenario, we will be optimizing the navigator parameters using SGD over mini-batches, and apply the navigators optimized with a mini-batch size of $N$ to calculate the conditional transport plans between the two discrete sets of $S = 500$ data points. More specifically, we use (14) as the loss to train the navigator parameter $\phi$ (note here both distributions are fixed and there is no generator to train), with the mini-batch size set as $N = M = 50, 200, 400,$ or 500. Each mini-batch consists of two sets of $M$ data points $iid$ drawn from their corresponding discrete distributions (*i.e.*, sampled without replacement from their corresponding discrete sets). With the navigator parameter $\phi$ optimized under the ACT divergence, the bottom panel of Fig. 9 reports the forward conditional transport plan from the Gaussian to $p_Y$, *i.e.*, $(\pi(y_1 \mid x, \phi), \ldots, \pi(y_S \mid x, \phi))$, where $x \sim \mathcal{N}(-1000, 1)$ and $\phi$ is learned with four different mini-batch sizes. There results suggest that even though for two discrete distributions whose supporting points contain outliers, there is a low risk for the conditional transport plans optimized under ACT to degenerate.

## B.3 MORE RESULTS ON 2D TOY DATASETS

We visualize the results on the 8-Gaussian mixture and three additional 2D toy datasets. Compared to the 8-Gaussian mixture dataset, the mode collapse issue of both GAN and WGAN-GP becomes more severe on the Swiss-Roll, Half-Moon, and 25-Gaussian datasets, while ACT consistently shows good and stable performance on all of them.

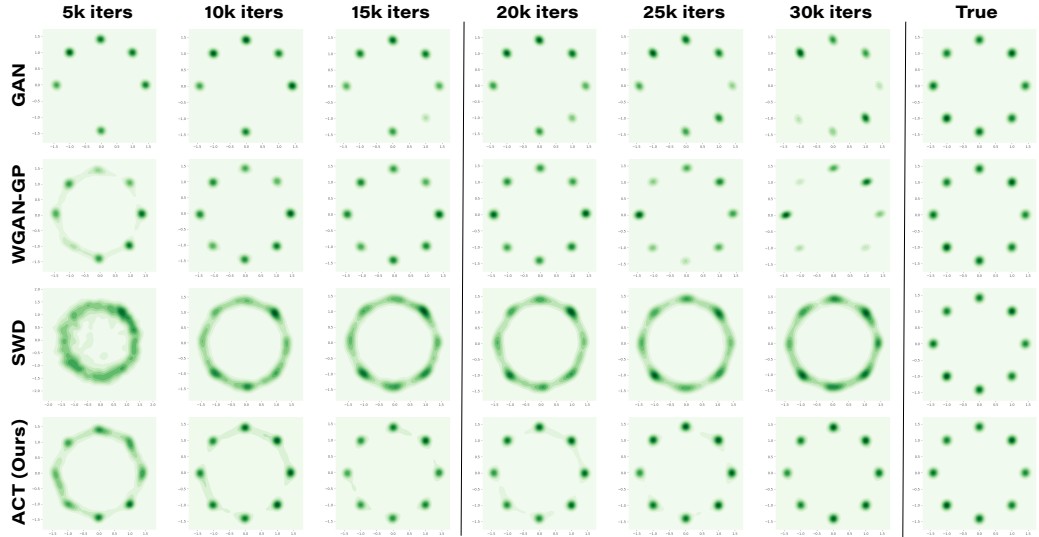

Figure 10: On a 8-Gaussian mixture data, comparison of generation quality and training stability between two mini-max deep generative models (DGMs), including vallina GAN and Wasserstein GAN with gradient penalty (WGAN-GP), and two mini-max-free DGMs, whose generators are trained under the sliced Wasserstein distance (SWD) and the proposed ACT divergence, respectively. The critics of GAN and WGAN-GP and the navigators of ACT are fixed after $15k$ iterations. The first column shows the true data density.

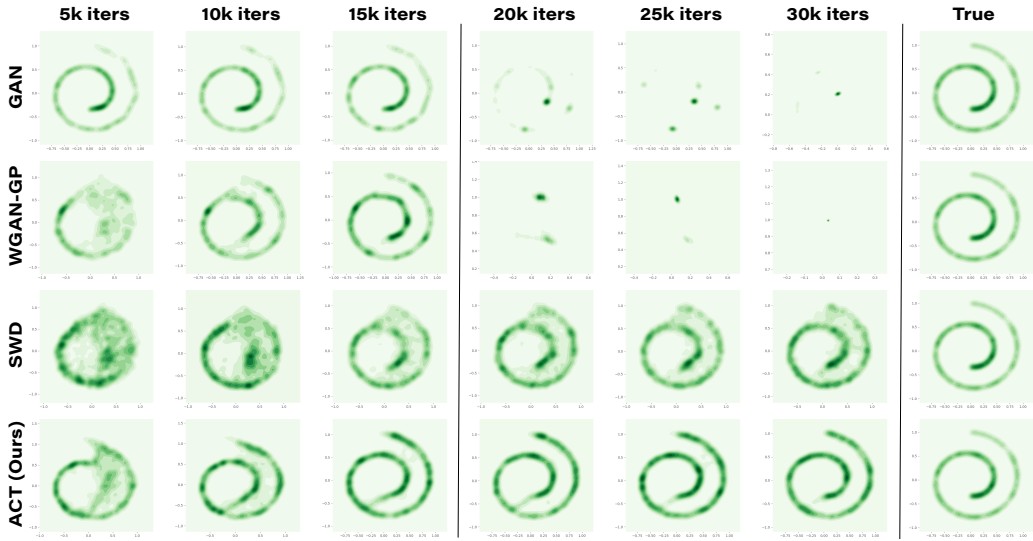

Figure 11: Analogous plot to Fig. 10 for the Swiss-Roll dataset.

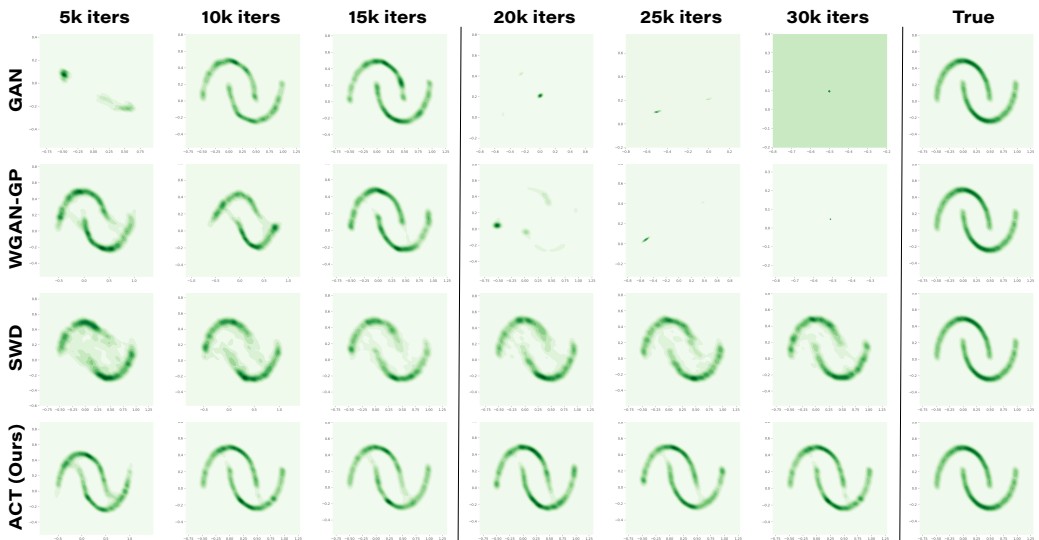

Figure 12: Analogous plot to Fig. 10 for the Half-Moon dataset.

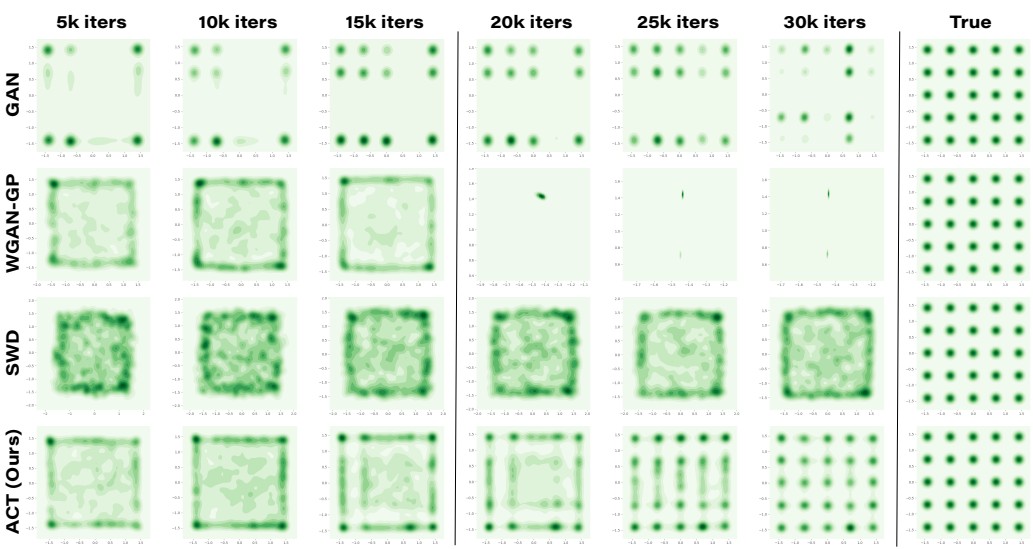

Figure 13: Analogous plot to Fig. 10 for the 25-Gaussian mixture dataset.

We also illustrate the data points and generated samples with empirical samples, shown in Fig. 14. The first column shows the generated samples (marked in blue) and the samples from data distribution (marked in red). To visualize how the feature extractor $\mathcal{T}_\phi$ used by both navigators works, we set its output dimension as 1 and plot the logits in the third and fifth columns and map the corresponding data (in the second column) and generated samples (in the fourth column) with the same color.

Similarly, we visualize the GAN's generated samples and logits produced by its discriminator in Fig. 15. We can observe that the discriminator maps the data to very close values. Specifically, in both the 8-Gaussian mixture and 25-Gaussian mixture cases, when the mode collapse occurs, the logits of the missed modes have similar value to the those in the other modes. This property results in GAN's mode collapse problem and it is commonly observed in GANs. Different from the GAN case, the navigator in our ACT model maps the data with non-saturating logits. We can observe in various multi-mode cases, different modes are assigned with different values by the navigator. This property helps ACT to well resist the mode collapse problem and stabilize the training.

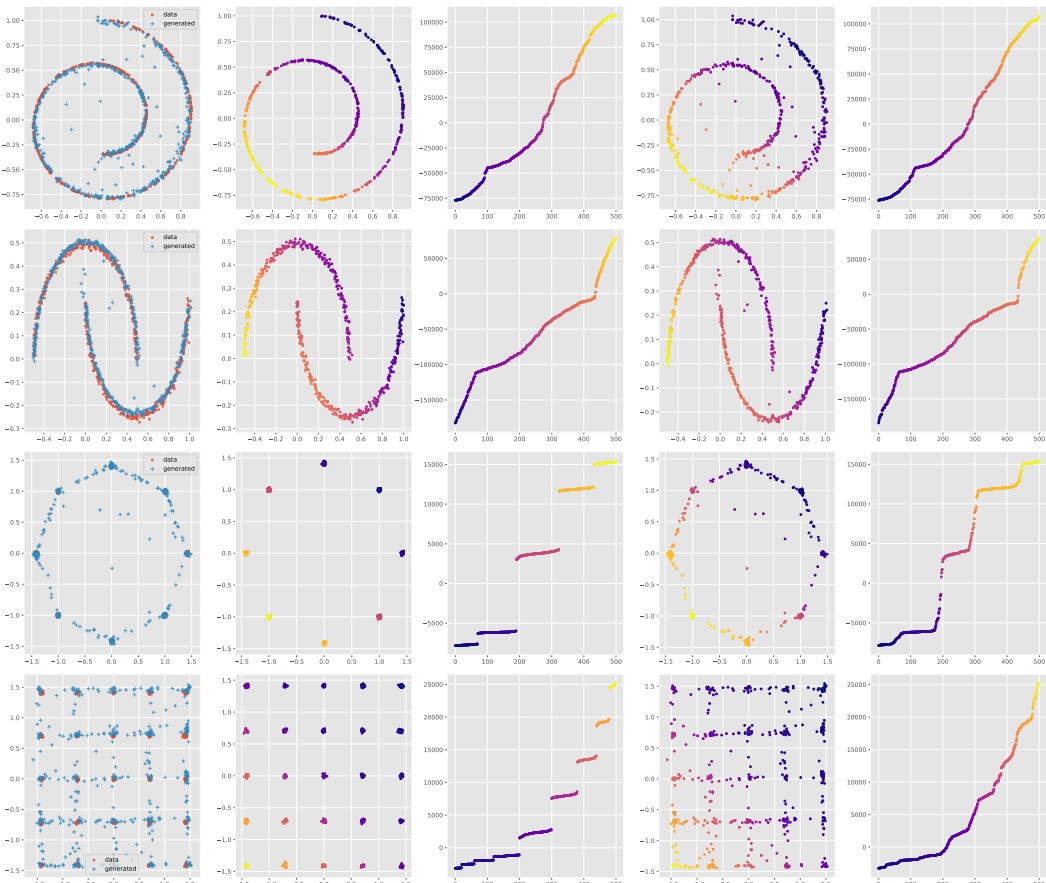

Figure 14: Visual results of *ACT* for generated samples (blue dots) compared to real samples (red dots) on Swiss Roll, Half Moons, 8-Gaussian mixture, and 25-Gaussian mixture. The second and third columns map the data points and their corresponding *navigator* logits by color; The fourth and fifth columns map the generated points and their corresponding *navigator* logits by color.

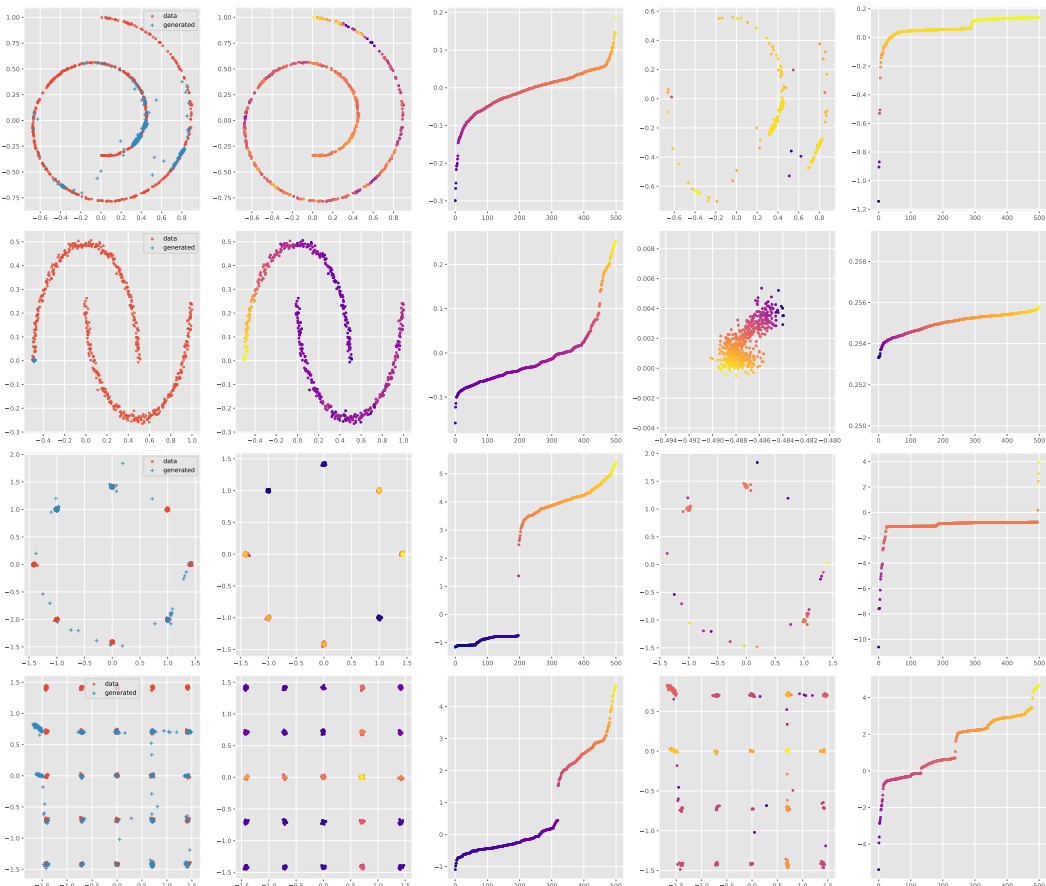

Figure 15: Visual results of *GAN* for generated samples (blue dots) compared to real samples (red dots) on Swiss Roll, Half Moons, 8-Gaussian mixture, and 25-Gaussian mixture. The second and the third columns map the data points and their corresponding *discriminator* logits by color; The fourth and fifth columns map the generated points and the corresponding *discriminator* logits by color.

### B.4 RESULTS FOR ABLATION STUDY

**Transport cost in pixel space *vs.* feature space**  We visualize the difference of using the transport cost in the pixel space and in the feature space here. In both Figs. 16 and 17, we test with MNSIT and CIFAR-10 data and with the $\mathcal{L}_2^2$ distance and cosine dissimilarity as the transport cost, respectively. For the MNIST dataset, due to its simple data structure, ACT can still be trained to generate meaningful digits, though some digits appear blurry. On the CIFAR-10, we can observe the model fails to generate any class of CIFAR images. As the dimensionality of the input space increases, using the distance in the pixel space as transport cost might lose the essential information for the transport and increases the training complexity of the navigator.

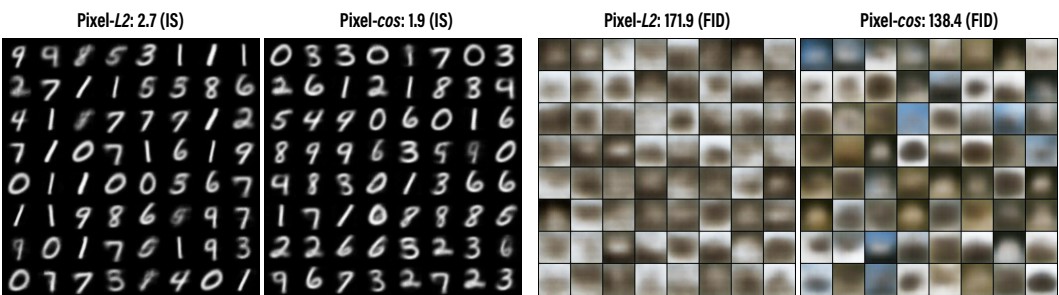

Figure 16: Visual results of generated samples on MNIST and CIFAR-10 using pixel-wise transport cost, with DCGAN (standard CNN) backbone.

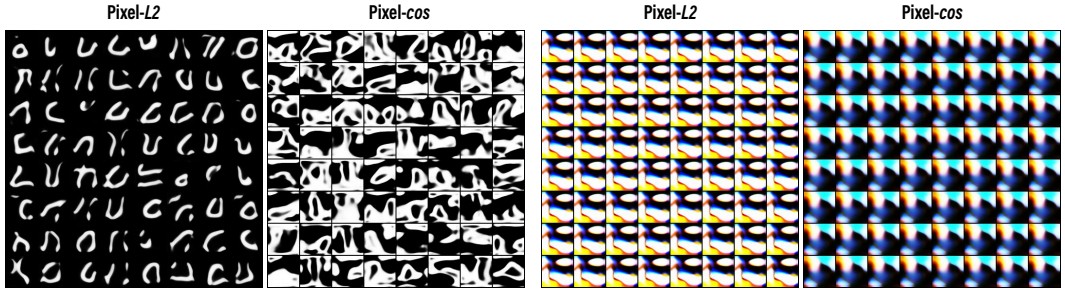

Figure 17: Visual results of generated samples on MNIST and CIFAR-10 using pixel-wise transport cost, with SNGAN (ResNet) backbone. The Inception and FID scores are not shown due to poor visual quality.

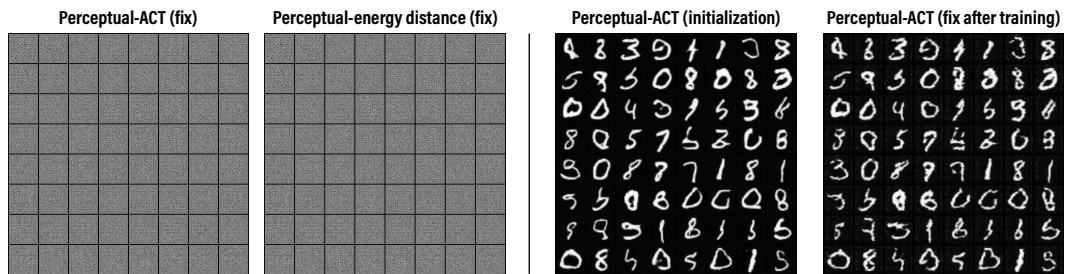

Figure 18: Visual results of generated samples with the perceptual similarity (Zhang et al., 2018) with four different training configurations.

We also consider using the perceptual similarity (Zhang et al., 2018) to define the cost function. Here we test with four configurations: 1) Apply a fixed and pre-trained perceptual loss to calculate the distance between the data and generated samples, use that distance as the point-to-point cost, and calculate ACT to train the generator; 2) Apply a fixed and pre-trained perceptual loss to calculate

Table 2: FID comparison for ACT-DCGAN and ACT-SNGAN on CIFAR-10 with different cost and architecture.

| **-Standard CNN-** | | transport cost $c(\boldsymbol{x}, \boldsymbol{y})$ | | | |
| | | $\mathcal{L}_2^2$-pixel | $cos$-pixel | $\mathcal{L}_2^2$-feature | $cos$-feature |
| Navigator cost | $\mathcal{L}_2^2$ | 171.9 | 138.4 | 27.1 | 24.8 |
| $d(\mathcal{T}_\phi(\boldsymbol{x}), \mathcal{T}_\phi(\boldsymbol{y}))$ | $cos$ | 157.9 | 136.2 | 25.3 | 25.0 |
| **-ResNet-** | | transport cost $c(\boldsymbol{x}, \boldsymbol{y})$ | | | |
| | | $\mathcal{L}_2^2$-pixel | $cos$-pixel | $\mathcal{L}_2^2$-feature | $cos$-feature |
| Navigator cost | $\mathcal{L}_2^2$ | 237.2 | 297.1 | 22.8 | 21.6 |
| $d(\mathcal{T}_\phi(\boldsymbol{x}), \mathcal{T}_\phi(\boldsymbol{y}))$ | $cos$ | 293.1 | 252.7 | 19.8 | 18.0 |

the energy distance between the data and generated samples to train the generator; 3) Apply the pre-trained perceptual loss as cost and fine-tune with ACT to train the generator; 4) Apply the pre-trained perceptual loss as cost, fine-tune it with ACT to train the generator, and then fix it for 20 more training epochs. We report the visual results of these 4 configurations in Fig. 18. As shown, fixing the metric and calculate the distance (either ACT or energy distance) in the feature space does not show good generation results. An explanation could be the pretrained perceptual loss is not trained on the generation task and hence might not be compatible with the learning objective. Thus the cost could not feedback useful signal to guide the generator. Using the energy distance as in Bellemare et al. (2017) shows similar results. We further use the perceptual network as the initialization of our critic $\mathcal{T}_\eta$, and train with ACT by maximizing the cost for 40 epochs on MNIST, which produce good-quality generations as shown in the third column. Then we fix the training of this critic and train the generator for 20 more epochs, which lead to degraded generation quality. We expect the generation quality will get worse as the training under the fixed critic continues.

**Using alternative cost function** We also test ACT with different configurations. As discussed in previous sections, the defined cost may also affect the training of the model. Here we vary the choice of transport cost $c_\eta(\boldsymbol{x}, \boldsymbol{y})$ and the navigator cost $d(\mathcal{T}_\phi(\boldsymbol{x}), \mathcal{T}_\phi(\boldsymbol{y}))$. For both cost, we test with $L_2^2$ distance and cosine dissimilarity. Moreover, we also compare the effects of distance in the original pixel space and the feature space (equipped with critic $\mathcal{T}_\eta$). The results in Table 2 highlight the importance of the cost in the feature space when dealing with high-dimensional image data. Moreover, compared to the $L_2^2$ distance, the cosine dissimilarity is observed to improve the model when applied as transport and navigator cost, especially with ResNet architecture.

**Training the critic with the discriminator loss of a vanilla GAN** Contrary to the existing critic-based GANs, the sample estimates of ACT divergence and its gradient are unbiased regardless of how well the critic is trained. We thus keep the same experiment settings and train the ACT-DCGAN model's critic with the discriminator loss of standard GANs, *i.e.* $\mathbb{E}_{x \sim \mathbb{P}_d}\left[-\log(\mathcal{T}_\eta(x)\right] + \mathbb{E}_{x \sim \mathbb{P}_g}\left[-\log(1 - \mathcal{T}_\eta(x))\right]$. The results in Fig. 19 shows that ACT works well in conjunction with the alternative critic training. The quantitative and qualitative on MNIST, CIFAR-10, CelebA, and LSUN are shown in Fig. 19. We can observe the quality of generated samples, while clearly not as good as training the critic with the ACT divergence, can still catch up with some of the benchmarks in Table 1.

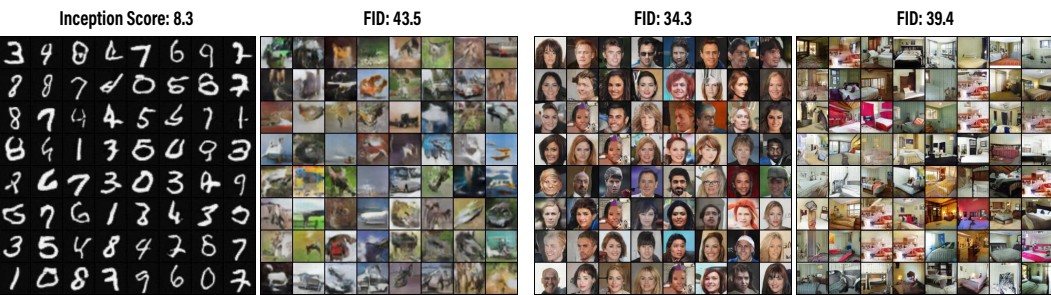

Figure 19: Visual results of using a standard cross-entropy discriminator loss in lieu of ACT divergence to train the critic of ACT.

### B.5 MORE RESULTS ON IMAGE DATASETS

For the experiments on the image datasets, we provide more visual results in this part. Apart from the datasets described in the experiment part, we also test the capacity of single-channel image generation with the MNIST dataset. Considering the inception score and the FID score are designed for RGB natural images, we also calculate the inception score of the real testing sets for reference. The presented methods are all able to generate meaningful digits on MNIST. If we take a closer look at the digits, the digits generated with $\mathcal{L}_2$ cost is less natural than the one with cosine cost. Moreover, we show both unconditional and conditional generation results on CIFAR-10. For both unconditional and conditional generation, our proposed method achieves good quantitative and qualitative results.

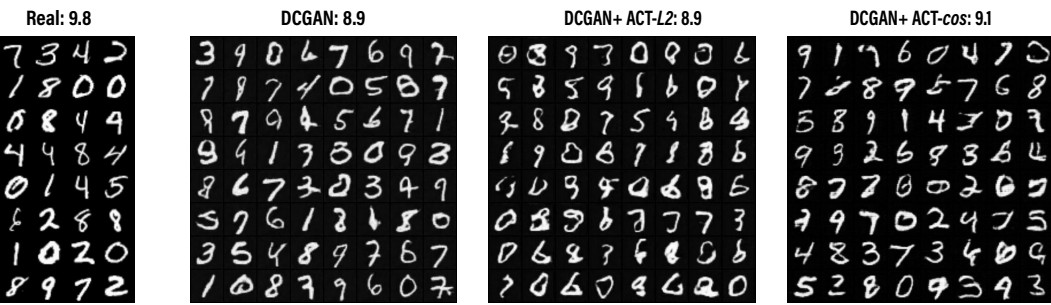

Figure 20: Unconditional generated samples and inception scores of MNIST, with DCGAN (standard CNN) backbone.

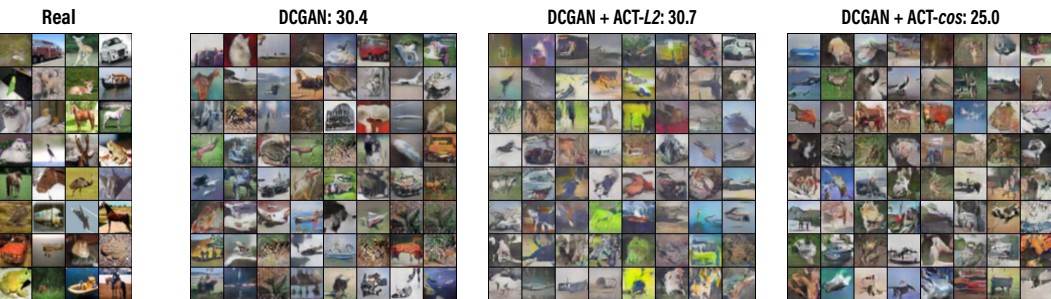

Figure 21: Unconditional generated samples and FIDs of CIFAR-10, with DCGAN (standard CNN) backbone.

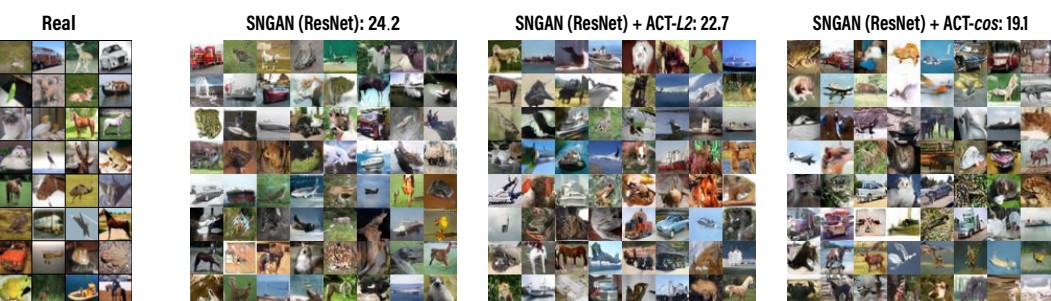

Figure 22: Unconditional generated samples and FIDs of CIFAR-10, with SNGAN (ResNet) backbone.

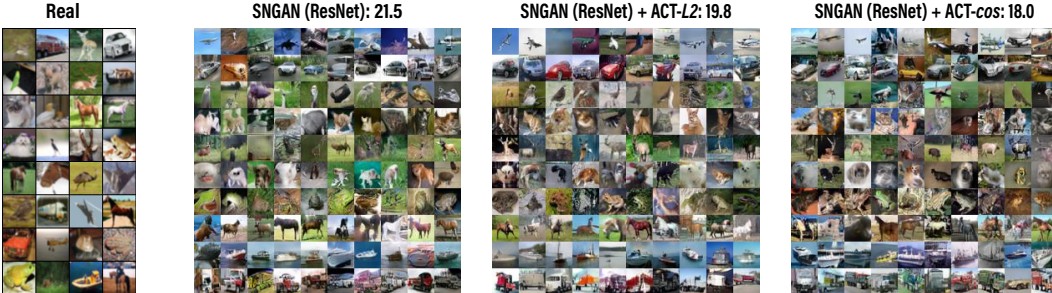

Figure 23: Conditional generated samples and FIDs of CIFAR-10, with SNGAN (ResNet) backbone.

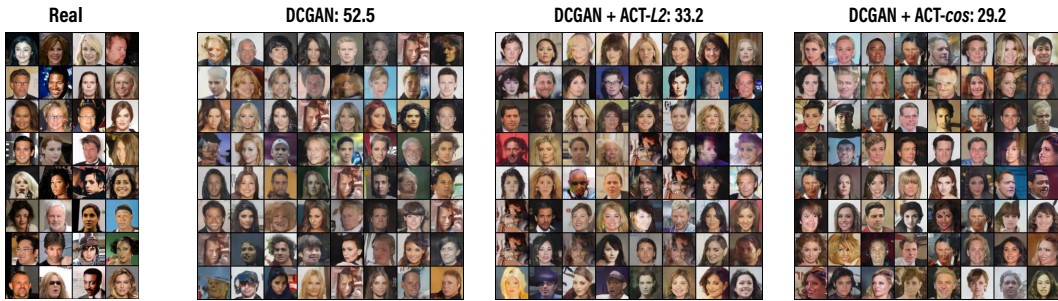

Figure 24: Generated samples and FIDs of CelebA, with DCGAN (standard CNN) backbone.

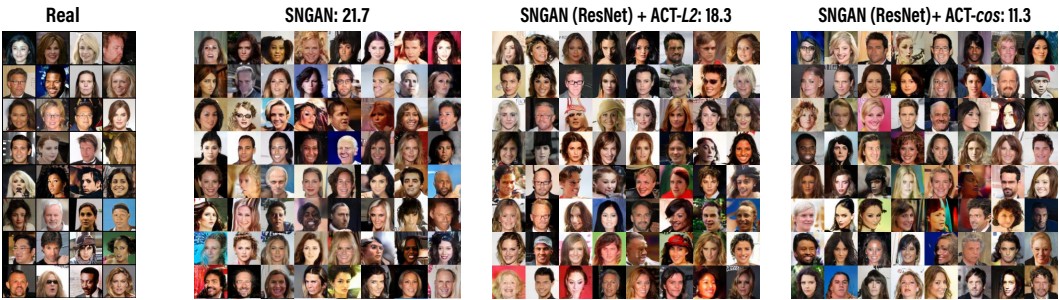

Figure 25: Generated samples and FIDs of CelebA, with SNGAN (ResNet) backbone.

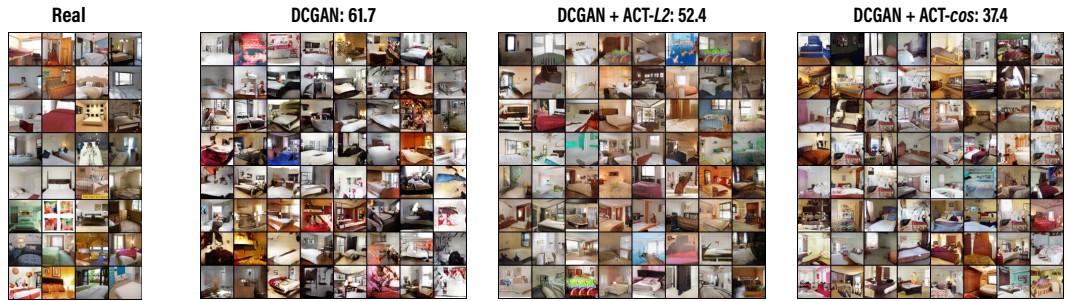

Figure 26: Generated samples and FIDs of LSUN, with DCGAN (standard CNN) backbone.

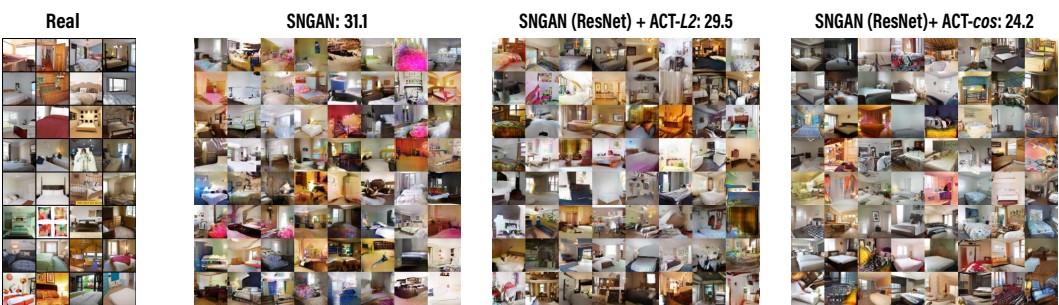

Figure 27: Generated samples and FIDs of LSUN, with SNGAN (ResNet) backbone.

## B.6 Experiment Details

**Preparation of datasets**  We apply the commonly used training set of MNIST (50K images, $28 \times 28$ pixels) (Lecun et al., 1998), CIFAR-10 (50K images, $32 \times 32$ pixels) (Krizhevsky et al., 2009), CelebA (about 203K images, resized to $64 \times 64$ pixels) (Liu et al., 2015), and LSUN bedrooms (around 3 million images, resized to $64 \times 64$ pixels) (Yu et al., 2015). The images were scaled to range $[-1, 1]$. For MNIST, when calculate the inception score, we repeat the channel to convert each gray-scale image into a RGB format.

**Network architecture and hyperparameters**  For the network architectures presented here, the slopes of all lReLU functions in the networks are set to 0.1 by default. For toy experiments, typically $10,000$ update steps are sufficient. However, our experiments show that the DGM optimized with the ACT divergence can be stably trained at least over $500,000$ steps (or possibly even more if allowed to running non-stop) regardless of whether the navigators are frozen or not after a certain number of iterations, where the GAN's discriminator usually diverges long before reaching that many iterations even if we do not freeze it after a certain number of iterations. For all image experiments, the output feature dimension of the navigators and that of the critic (*i.e.*, $\mathcal{T}_{\boldsymbol{\phi}}(\cdot), \mathcal{T}_{\boldsymbol{\eta}}(\cdot) \in \mathbb{R}^m$) are set to $m = 2048$. All models are able to be trained on a single GPU, such as NVidia GTX 1080-TI in our experiments, with $150,000$ generator updates (for CIFAR-10 we apply $50,000$ iterations).

To keep close to the configuration of the DCGAN and SNGAN experiments setting, we use the the Adam optimizer (Kingma and Ba, 2015) with learning rate $\alpha = 2 \times 10^{-4}$ and $\beta_1 = 0.5$, $\beta_2 = 0.99$ for the parameters of the generator, navigators, and critic. On the DCGAN backbone, we let all the modules update with the same frequency; while on the SNGAN backbone, the critic is updated once per 5 generator updating steps. The performance might be further improved with more careful fine-tuning. For example, the learning rate of the navigator parameter could be made smaller than that of the generator parameter. The true data minibatch size is fixed to $N = 64$ for all experiments. Moreover, with this batch-size we let the generated sample size $M$ the same as minibatch size $N$ for ACT computation and we have monitored the average time for each update step on a single NVidia GTX 1080-TI GPU: On CIFAR-10, each update step takes around 0.1s and 0.2s for DCGAN and SNGAN, respectively; For DCGAN and SNGAN backbone trained with ACT divergence each update step takes around 0.4s and 0.7s. On CelebA and LSUN, each update takes 0.6s and 0.7s for DCGAN and SNGAN, respectively; when trained with ACT, the elapsed time for each update increases to 3.3s and 3.6s, respectively.

Table 3: Network architecture for toy datasets ($V$ indicates the dimensionality of data).

| (a) Generator $G_{\boldsymbol{\theta}}$ |
| :---: |
| $\boldsymbol{\epsilon} \in \mathbb{R}^{50} \sim \mathcal{N}(0, 1)$ |
| $50 \rightarrow 100$, dense, lReLU |
| $100 \rightarrow 50$, dense, lReLU |
| $50 \rightarrow V$, dense, linear |

| (b) Navigator $\mathcal{T}_{\boldsymbol{\phi}}$ / Discriminator $D_{\boldsymbol{\phi}}$ |
| :---: |
| $\boldsymbol{x} \in \mathbb{R}^V$ |
| $2 \rightarrow 100$, dense, lReLU |
| $100 \rightarrow 50$, dense, lReLU |
| $50 \rightarrow 1$, dense, linear |

Table 4: DCGAN architecture for the CIFAR-10 dataset ($h = w = 4$).

(a) Generator $G_{\boldsymbol{\theta}}$

| $\boldsymbol{\epsilon} \in \mathbb{R}^{128} \sim \mathcal{N}(0,1)$ |
| --- |
| $128 \to 4 \times 4 \times 512$, dense, linear |
| $4 \times 4$, stride=2 deconv. BN 256 ReLU |
| $4 \times 4$, stride=2 deconv. BN 128 ReLU |
| $4 \times 4$, stride=2 deconv. BN 64 ReLU |
| $3 \times 3$, stride=1 conv. 3 Tanh |

(b) Navigator $\mathcal{T}_{\boldsymbol{\phi}}$ / Critic $\mathcal{T}_{\boldsymbol{\eta}}$

| $\boldsymbol{x} \in [-1,1]^{32 \times 32 \times 3}$ |
| --- |
| $3 \times 3$, stride=1 conv 64 lReLU
$4 \times 4$, stride=2 conv 64 lReLU |
| $3 \times 3$, stride=1 conv 128 lReLU
$4 \times 4$, stride=2 conv 128 lReLU |
| $3 \times 3$, stride=1 conv 256 lReLU
$4 \times 4$, stride=2 conv 256 lReLU |
| $3 \times 3$, stride=1 conv. 512 lReLU |
| $h \times w \times 512 \to m$, dense, linear |

Table 5: DCGAN architecture for the CelebA and LSUN datasets ($h = w = 4$).

(a) Generator $G_{\boldsymbol{\theta}}$

| $\boldsymbol{\epsilon} \in \mathbb{R}^{128} \sim \mathcal{N}(0,1)$ |
| --- |
| $128 \to 4 \times 4 \times 1024$, dense, linear |
| $4 \times 4$, stride=2 deconv. BN 512 ReLU |
| $4 \times 4$, stride=2 deconv. BN 256 ReLU |
| $4 \times 4$, stride=2 deconv. BN 128 ReLU |
| $4 \times 4$, stride=2 deconv. BN 64 ReLU |
| $3 \times 3$, stride=1 conv. 3 Tanh |

(b) Navigator $\mathcal{T}_{\boldsymbol{\phi}}$ / Critic $\mathcal{T}_{\boldsymbol{\eta}}$

| $\boldsymbol{x} \in [-1,1]^{64 \times 64 \times 3}$ |
| --- |
| $4 \times 4$, stride=2 conv 64 lReLU
$4 \times 4$, stride=2 conv BN 128 lReLU |
| $4 \times 4$, stride=2 conv BN 256 lReLU |
| $3 \times 3$, stride=1 conv BN 512 lReLU |
| $h \times w \times 512 \to m$, dense, linear |

Table 6: ResNet architecture for the CIFAR-10 dataset.

(a) Generator $G_{\boldsymbol{\theta}}$

| $\boldsymbol{\epsilon} \in \mathbb{R}^{128} \sim \mathcal{N}(0,1)$ |
| --- |
| $128 \to 4 \times 4 \times 256$, dense, linear |
| ResBlock up 256 |
| ResBlock up 256 |
| ResBlock up 256 |
| BN, ReLU, $3 \times 3$ conv, 3 Tanh |

(b) Navigator $\mathcal{T}_{\boldsymbol{\phi}}$ / Critic $\mathcal{T}_{\boldsymbol{\eta}}$

| $\boldsymbol{x} \in [-1,1]^{32 \times 32 \times 3}$ |
| --- |
| ResBlock down 128 |
| ResBlock down 128 |
| ResBlock 128 |
| ResBlock 128 |
| ReLU |
| Global sum pooling |
| $h = 128 \to m$, dense, linear |

Table 7: ResNet architecture for the CelebA and LSUN datasets.

(a) Generator $G_{\boldsymbol{\theta}}$

| |
|---|
| $\boldsymbol{\epsilon} \in \mathbb{R}^{128} \sim \mathcal{N}(0,1)$ |
| $128 \to 4 \times 4 \times 1024$, dense, linear |
| ResBlock up 512 |
| ResBlock up 256 |
| ResBlock up 128 |
| ResBlock up 64 |
| BN, ReLU, $3 \times 3$ conv, 3 Tanh |

(b) Navigator $\mathcal{T}_{\boldsymbol{\phi}}$ / Critic $\mathcal{T}_{\boldsymbol{\eta}}$

| |
|---|
| $\boldsymbol{x} \in [-1,1]^{64 \times 64 \times 3}$ |
| ResBlock down 128 |
| ResBlock down 256 |
| ResBlock down 512 |
| ResBlock down 1024 |
| ReLU 
 Global sum pooling |
| $h = 1024 \to m$, dense, linear |

Table 8: ResNet architecture for the LSUN-128 dataset.

(a) Generator $G_{\boldsymbol{\theta}}$

| |
|---|
| $\boldsymbol{\epsilon} \in \mathbb{R}^{128} \sim \mathcal{N}(0,1)$ |
| $128 \to 4 \times 4 \times 1024$, dense, linear |
| ResBlock up 1024 |
| ResBlock up 512 |
| ResBlock up 256 |
| ResBlock up 128 |
| ResBlock up 64 |
| BN, ReLU, $3 \times 3$ conv, 3 Tanh |

(b) Navigator $\mathcal{T}_{\boldsymbol{\phi}}$ / Critic $\mathcal{T}_{\boldsymbol{\eta}}$

| |
|---|
| $\boldsymbol{x} \in [-1,1]^{64 \times 64 \times 3}$ |
| ResBlock down 128 |
| ResBlock down 256 |
| ResBlock down 512 |
| ResBlock down 1024 |
| ResBlock 1024 |
| ReLU 
 Global sum pooling |
| $h = 1024 \to m$, dense, linear |

Table 9: ResNet architecture for the CelebA-HQ dataset.

(a) Generator $G_{\boldsymbol{\theta}}$

| |
|---|
| $\boldsymbol{\epsilon} \in \mathbb{R}^{128} \sim \mathcal{N}(0,1)$ |
| $128 \to 4 \times 4 \times 1024$, dense, linear |
| ResBlock up 1024 |
| ResBlock up 512 |
| ResBlock up 512 |
| ResBlock up 256 |
| ResBlock up 128 |
| ResBlock up 64 |
| BN, ReLU, $3 \times 3$ conv, 3 Tanh |

(b) Navigator $\mathcal{T}_{\boldsymbol{\phi}}$ / Critic $\mathcal{T}_{\boldsymbol{\eta}}$

| |
|---|
| $\boldsymbol{x} \in [-1,1]^{64 \times 64 \times 3}$ |
| ResBlock down 128 |
| ResBlock down 256 |
| ResBlock down 512 |
| ResBlock down 512 |
| ResBlock down 1024 |
| ResBlock 1024 |
| ReLU 
 Global sum pooling |
| $h = 1024 \to m$, dense, linear |

