# OpenReview forum: "ACT: Asymptotic Conditional Transport"
_ICLR.cc/2021/Conference — Reject_

### Official Review · AnonReviewer4 · 2020-10-28
**A new divergence between probability distributions and its potential application in GANs (replacing Wasserstein)**

**Rating:** 6
**Confidence:** 3

**Review:**

The paper proposes a new transport-based divergence between distributions (CT) and a variant for empirical distributions (ACT). The new divergence is claimed to be more suitable for learning deep generative models than existing divergences like KL, JS (as in the vanilla GAN) and Wasserstein (as used in WGAN and its variants).  The proposed divergence mostly resembles, in my opinion, the Wasserstein divergence variant that uses the  Kantorovich–Rubinstein dual definition (which requires the learned function to be 1-Lipschitz). It seems that the main advantages of ACT over Wasserstein is that there is no constraint on the Lipschitz smoothness (which has to be enforced in WGAN by means of e.g. gradient clipping or gradient penalty), and the fact that ACT provides unbiased gradients that do not require the critic to reach an optimal point (as required in theory in GAN or WGAN).

The paper presents a potentially interesting and significant method, however, it was a bit hard to follow, making it difficult for me to asses the actual significance of the contribution.

Specifically, the distinction between $d(x,y)$ and $c(x,y)$ was not 100% clear to me. $d(x,y)$ seems to be part of the "navigator" $\pi(x | y)$ - an energy-based conditional probability and $c(x, y)$ is the point-to-point transport cost. For the CT to be low, it looks like $c(x,y)$ and $d(x,y)$ should be positively correlated. It was not clear, however (at least from the introduction), how $d$ and $c$ fit the generator-critic GAN scheme. Are they both part of the critic? Do the claims that the critic does not have to reach optimality for the gradients to be unbiased refer just to $c$ or to $d$ as well? I believe this is partially explained, but only towards the end of the paper (eq. 16). Perhaps a figure showing the trained elements in ACT compared to the trained elements in WGAN could help clarify the method.

Specific questions:
- Why are both the forward and backward CT needed?
- Are the parameters of the forward and backward navigators shared? From eq. 16 it looks like they are. Does it make sense?
- In section 2.3, the authors claim that L2 distance in the original image domain is known not to work well for high-dimensional data that resided on a lower dimensional manifold, but what about L2 on some pre-trained feature-space e.g. perceptual distance [1]?
- The difference in principle between the proposed ACT and OT-GAN and MMD-GAN should be described.

To summarize:

pros:
- Interesting method, seems to be novel
- Potential significance (as an alternative loss for training deep generative models)

cons:
- Writing hard to follow
- Some open questions **update: the authors have answered most of my questions**

[1] Zhang, Richard, et al. "The unreasonable effectiveness of deep features as a perceptual metric." Proceedings of the IEEE conference on computer vision and pattern recognition. 2018.

---

> ### Author Response · Authors · 2020-11-21
> **Point-by-point response to AnonReviewer4**
>
> Thank you for your constructive comments and insightful questions. Below please find our point-by-point response.
>
> > Q1: “ Why are both the forward and backward CT needed?”
>
> Response: We have added an analysis to compare the forward and backward transport properties to help the readers to understand why ACT can well fit the distribution and not miss modes (see newly added paragraph in Page 8 and Figs. 5 & 8). As a result, we show that the forward conditional transport cost encourages the generator to cover all modes, while the backward conditional transport cost encourages it to seek a mode. To this end, the navigators, although introduced as additional components, do not make the training harder, but help to make the training more stable and converge to better results.
>
> If only using forward CT/ACT, we observe mode covering behavior, while if only using backward CT/ACT, we observe mode seeking behavior. Combining both of them strikes a good balance between mode covering and seeking, providing strong resistance to mode collapse.
>
> > Q2: “ Superficially, the distinction between d(x,y) and c(x,y) was not 100% clear to me. d(x,y) seems to be part of the "navigator" \pi(x | y) - an energy-based conditional probability and c(x, y) is the point-to-point transport cost. For the CT to be low, it looks like c(x,y) and d(x,y) should be positively correlated.”
>
> Response: We’d like to clarify that they are not necessarily positively correlated when the navigator parameter is different from the critic parameter, i.e., when $\phi \neq \eta$. A good example is shown in Figure 14 of Appendix B.3, where we show the generated data and corresponding logits. In this 2D toy data case, $c(x,y)$ is the Euclidean distance between point $x$ and $y$; while $d(x,y)$ is the Euclidean distance between $T_\phi(x)$ and $T_\phi(y)$. Let’s use the third row (8-Gaussian mixture) as an example, in the second and third panels, we use different colors to indicate the correspondence between each point $x$ and its $T_\phi(x)$. For the points from the two modes that are marked in pink, we can observe in the data space, they have the largest distance,  while in the navigator space, they are close to each other. The navigator space can be viewed as a 45-degree zero-crossing diagonal line in the figure, and hence the distances between the points in the navigator space are determined by their projections to this diagonal line, which also justifies that $c(x,y)$ and $d(x,y)$ are not necessarily positively correlated.
>
> > Q3: “ Are the parameters of the forward and backward navigators shared? From eq. 16 it looks like they are. Does it make sense?”
>
> Response: Yes, as defined in Eq. 2 and Eq. 3, they are shared.  While in the proposed construction, both conditional transport plans are defined as conditional distributions parameterized by the same $\phi$, this is not mandatory as long as the conditional distributions are valid and can be optimized. There could exist various alternative ways. For example, one may use one neural network for the forward navigator and another different one for the backward, which could help further increase flexibility and hence the performance, but at the expense of doubling the navigator-related memory and computation cost. We will further investigate this in our future work.
>
> > Q4: “ In section 2.3, the authors claim that L2 distance in the original image domain is known not to work well for high-dimensional data that resided on a lower-dimensional manifold, but what about L2 on some pre-trained feature-space e.g. perceptual distance.”
>
> Response: We have added a paragraph in Pages 22-23 and Fig. 18 to help answer this question.
> To summarize, our experiments do not show promising results for high-dimensional natural images, when fixing the pre-trained feature extractor and letting it play the role of the critic in ACT. Specifically, we apply the pre-trained feature extractor from Zhang et al. (2018) to define the critic of the ACT, but the results are not promising (Fig. 18) unless we allow the feature-extract network to be fine-tuned under the ACT loss shown in Eq. 16. We have also tested with other generative models such as energy distance, and none of them show good results either when the pre-trained feature extractor is used to define the critic and fixed during the training.

---

> > ### Author Response · Authors · 2020-11-21
> > **Point-by-point response to AnonReviewer4 (continue)**
> >
> > > Q5: “The difference in principle between the proposed ACT and OT-GAN and MMD-GAN should be described.”
> >
> > Response: We have clarified their difference in the revised paper (see the text highlighted in blue in Pages 3 and 4). In more detail, most of the OT-based GANs employ the Kantorovich duality to derive a min-max objective. Thus they are optimized just like the standard mini-max framework of vanilla GANs, but imposed with a Lipschitz constraint on the critic/discriminator. Some other OT-GANs apply the primal form and use the Sinkhorn distance for the optimization, which uses an entropic regularization for the transport plan. As the entropic regularization coefficient goes to 0,  we recover the OT distance and as it goes to infinity, we recover MMD. The MMD-GAN uses MMD discrepancy as the metric, which is evaluated in the kernel space that is learned by an adversarial mechanism. Moreover, if we apply the dual form of OT and constrain the critic in a certain space, we could also recover MMD. Like in Algorithm 1 in MMD-GAN [Li et al., 2017], they constrain the critic to be locally Lipschitz and apply weight clipping. Like neither OT-GAN nor MMD-GAN, ACT is not proposed to approximate the OT solution, but is somewhat related to an amortized version of the OT in its primal form. The properties of ACT ensure that its stochastic gradients stay unbiased on mini-batch based stochastic optimization. Moreover, it requires no Lipschitz constraint and hence avoids the need of using heuristics, such as weight clipping, to satisfy that constraint.

---

### Official Review · AnonReviewer1 · 2020-10-28
**request deeper studies on the proposed method's properties**

**Rating:** 6
**Confidence:** 4

**Review:**

Summary:

The paper introduces an asymptotic conditional transport divergence to measure the discrepancy between two probability distributions. The new divergence leads to a new adversarial game of generative adversarial networks, which aims at minimizing a distribution-to-distribution transport cost by optimizing the generator distribution and the conditional transport-path distributions of navigators.

Strength:

Introducing the asymptotic conditional transport divergence to generative modeling looks novel and interesting to me.
Both the motivation and the technical details seem sound.
The evaluation shows the effectiveness of the proposed method on both toy datasets and some popular image generation datasets.

Weakness:

Fig.3 has shown the superiority of the proposed divergence against the existing divergence (JS, Wasserstein distance, and Sliced Wasserstein distance) to fit the toy data distribution. But readers might doubt whether the comparison is fair or not. Do they use the same network backbone (e.g., DCGAN, WGAN-GP)? As there are many improved variants of SWD like (Deshpande et al., 2018, Deshpande et al., 2019, Wu et al., 2019), I am also wondering which SWD is used. The problem might get more serious for general cases. To address the problem, the paper is suggested to additionally present a clear theoretical study on the comparison, as readers really want to know why the proposed divergence works better than others, while some explanations have been presented in the introduction part.  For instance, it is not clear to me why the sample estimate of the ACT divergence and its gradient stay unbiased.

As the new divergence requests two additional navigators for bidirectional distribution-to-distribution transport, the adversarial game gets more complex in theory. It is known that GAN models generally suffer from unstable training and mode collapse issues. I am afraid the higher complexity might make such issues more serious. While Fig.1, Fig.2 somehow shows the behaviors of the navigators, I suggest to further study whether the proposed method can overcome or avoid such issues.

---

> ### Author Response · Authors · 2020-11-21
> **point-by-point response to AnonReviewer1**
>
> Thank you for your valuable comments and suggestions. Below please find our point-by-point response.
>
> > Q1: “ Fig.3 has shown the superiority of the proposed divergence against the existing divergence (JS, Wasserstein distance, and Sliced Wasserstein distance) to fit the toy data distribution. But readers might doubt whether the comparison is fair or not. Do they use the same network backbone (e.g., DCGAN, WGAN-GP)?”  “Which SWD is used?”
>
> Response: Yes, to ensure a fair comparison, on all toy data, we used the same architecture shown in Table 3 of Appendix B.6. Specifically, we applied the architecture in Table 3(a) for the generator of all the compared models, and applied the architecture in Table 3(b) for the discriminator of both GAN and WGAN-GP as well as the navigator of ACT, which means these models only differ in the training loss.  As described in the “ACT for 2D toy data” paragraph in Page 7, the SWD we used for toy data is from Deshpande et al., 2018. In Table 1, we included several additional SWD variants for comparison.
>
> > Q2: “ The paper is suggested to additionally present a clear theoretical study on the comparison, as readers really want to know why the proposed divergence works better than others.”
>
> We have added theoretical analysis and corresponding experiment results (see Fig. 3 in the revised paper) to show the advantage of ACT over the Wasserstein distance on mini-batch SGD based optimization. From our experiments, we can observe ACT consistently show good fitting results, which indicates ACT does not suffer the bias issue caused by the usage of mini-batches as the sample Wasserstein distance does.
>
>
> Theoretically, 1) if the objective is $\mathcal W(\mu,\nu)$, then the sample Wasserstein defined as $\mathcal W(\hat\mu_N,\hat\nu_M)$ is a biased estimator and its gradient is certainly biased. 2) If the objective is the expected sample Wasserstein defined as $E_{\hat\mu_N,\hat\nu_M}[ \mathcal W(\hat\mu_N,\hat\nu_M) ]$, then the sample Wasserstein $\mathcal W(\hat\mu_N,\hat\nu_M)$ is an unbiased estimator, computing which, however, requires solving a separate combinatorial problem for each mini-batch. 3) If the objective is $\mathcal W(\mu,\nu)$ and we estimate it using the dual form, then the gradient is unbiased only if the critic has reached its optimum.
>
> By contrast, our objective is the ACT divergence. As shown in Lemma 4, Eq (13) is an unbiased estimator and its gradient is unbiased. The usage of the navigators, whose parameter is optimized with SGD across mini-batches, amortizes the computation of the transport plans on each mini-batch.
>
>
> > Q3: “ It is known that GAN models generally suffer from unstable training and mode collapse issues. I am afraid the higher complexity might make such issues more serious.  While Fig.1, Fig.2 somehow shows the behaviors of the navigators, I suggest to further study whether the proposed method can overcome or avoid such issues.”
>
> We have followed your suggestion to conduct additional experiments to study whether ACT can effectively overcome these well-known issues of GANs. In Fig. 4 of the revised paper, we have added two experiments to show how ACT resists the mode collapse problem. We have also added an analysis of the amortization property compared to optimal transport (please refer to our response to Q1 of Reviewer3). Moreover, we have provided more analysis on the distinction between the forward and backward transport (please refer to our response to Q1 of Reviewer4) to further justify how ACT resists mode collapse issues.

---

### Official Review · AnonReviewer3 · 2020-11-08
**interesting idea but needs more experiments**

**Rating:** 5
**Confidence:** 4

**Review:**

The paper proposes conditional transport as a new divergence to measure the difference between two distributions. The idea is to learn the conditional transport plan of transporting one point in one distribution to the other marginal distribution. This conditional transport plan is modeled using a neural network. The resulting model is then applied to optimal transport formulation. Experiments are shown on image-based generative modeling dataset.

The main idea is to decompose the joint transportation plan $\pi(x, y)$ using conditionals as $p(x)\pi(y|x)$ and $p(y)\pi(x|y)$. The conditional transportation plan $\pi(x|y)$ and $\pi(y|x)$ are then modeled using neural contrastive losses, with the idea being similar points in two distributions are mapped closer. This decomposition is then applied in optimal transport formulation, leading to a new objective for optimizing OT. The authors also derive an empirical version of this objective, that makes it amenable for stochastic mini-batch optimization.

While the idea of this decomposition is interesting, I see the following issues with the formulation.

(1) From the forms of conditional transportation plans $\pi(x|y)$ and $\pi(y|x)$ in Eq 2 and 3, $p(x)\pi(y|x) \neq p(y)\pi(x|y)$. It would be good to have models that satisfy this equality.

(2) I am not sure if the contrastive model assumed in Eq 2 and 3 is expressive enough to model the entire space of marginal constraints in OT. That is, I don’t think $p(x)\pi(y|x)$ will cover $\Pi(\mu, \nu)$. Lemma 1 just shows that under some conditions, $p(x)\pi(y|x)$ lies inside $\Pi(\mu, \nu)$, but it would be interesting to see if then entire space of $\Pi(\mu, \nu)$ is covered by the neural contrastive model.

(3) When critic is used in ground cost function $c(x, y)$, the resulting optimization (Eq 16) has one additional network compared to standard GANs. Also, the adversarial game still exists. So, it looks like this optimization is more harder (or at least equally harder) compared to standard GANs. Is this true? Do you see any optimization benefits compared to standard GANs?

(4) The results of image-based generative modeling is not that impressive. On CIFAR and LSUN datasets, the performance is similar / slightly better than the compared GAN models. Also, many SOTA GAN models are not compared. So, it is very hard to say if the proposed model advances SOTA results.

(5) I would have liked to see more interesting results. The formulation of authors gives an estimate of transportation plan $\pi(x, y)$ in addition to the generative model itself, which is not possible to estimate in dual-based OT GAN optimization. The transportation plan can be used in interesting applications. One possibility is to estimate likelihoods and possibly use in OOD detection. Take a look at Balaji et al., “Entropic GANs meet VAEs: A statistical approach to compute sample likelihoods in GANs” for this. This is just one idea. Other interesting applications could have been demonstrated as well.

Overall, while the idea is interesting, more results could have positioned the paper better. Just selling it as a paper that improves image-based generative modeling is not that great in my opinion. I would encourage authors to think more interesting experiments.

---

> ### Author Response · Authors · 2020-11-11
> **The proposed conditional transport imposes less constraint on the joint probability measure than optimal transport does**
>
> Dear AnonReviewer3,  we’d like to first provide a response to your first two concerns, clarifying which is important to help understand how the proposed conditional transport (CT) differs from optimal transport.
>
> Concern (1): From the forms of conditional transportation plans $\pi(x|y)$ and and $\pi(y|x)$ in Eq 2 and 3, $p(x)\pi(y|x)\neq p(y)\pi(x|y)$ . It would be good to have models that satisfy this equality.
>
> Our response: It is actually our intention to not imposing this equality. An important feature of the proposed CT is to allow $p(x)\pi(y|x)\neq p(y)\pi(x|y)$, which makes it differ from optimal transport in imposing less constraint on the joint distribution of $x$ and $y$. To be more specific, denoting $\pi(x,y)=p(x)\pi(y|x)$, if $p(x)\pi(y|x) = p(y)\pi(x|y)$ is enforced, then
> $$\int \pi(x,y)dy = \int p(x)\pi(y|x)dy = p(x)\int \pi(y|x)dy=p(x)$$
> $$\int \pi(x,y)dx = \int p(y)\pi(x|y)dx =p(y)\int \pi(x|y)dx = p(y),$$
> which means that the joint probability measure $\pi \in \Pi(\mu,\nu)$. Thus we will have  $$E_{x\sim p(x)}E_{y\sim \pi(y|x)}[c(x,y)] = E_{y\sim p(y)}E_{x\sim \pi(x|y)}[c(x,y)] =E_{(x,y)\sim \pi(x,y)}[c(x,y)]$$
> and hence finding the best $\pi(y|x)$ (or $\pi(x|y)$) that minimizes the above expectation, subject to the constraint $p(x)\pi(y|x) = p(y)\pi(x|y)$, is the same as the optimal transport problem that finds the best $\pi(x,y)$ to minimize the above expectation subject to the constraint $\pi \in \Pi(\mu,\nu)$.
>
> In the proposed CT, we do not impose $p(x)\pi(y|x)= p(y)\pi(x|y)$.  The cost of the forward CT is defined as $E_{x\sim p(x)}E_{y\sim \pi(y|x)}[c(x,y)]$, where the joint distribution of $x$ and $y$, denoted as $\pi_{forward}(x,y) =p(x)\pi(y|x)$, satisfies $\int \pi_{forward}(x,y) dy=p(x)$ but is not constrained to satisfy $\int \pi_{forward}(x,y) dx=p(y)$. Similarly, the joint distribution in the backward CT, denoted as $\pi_{backward}(x,y) =p(y)\pi(x|y)$, satisfies $\int \pi_{backward}(x,y) dx=p(y)$ but is not constrained to satisfy $\int \pi_{forward}(x,y) dy=p(x)$. Therefore, in theory we have $$\min_{\pi(y|x)} E_{(x,y)\sim\pi_{forward}(x,y) }[c(x,y)]\le \min_{\pi\in \Pi(\mu,\nu)} E_{(x,y)\sim\pi(x,y) }[c(x,y)] $$
> and
> $$\min_{\pi(x|y)}E_{(x,y)\sim\pi_{backward}(x,y) }[c(x,y)]\le \min_{\pi\in \Pi(\mu,\nu)} E_{(x,y)\sim\pi(x,y) }[c(x,y)] $$
> which means the smallest possible cost of the (forward/backward) CT is smaller than or equal to the Wasserstein distance.
>
>
> Concern (2): I am not sure if the contrastive model assumed in Eq 2 and 3 is expressive enough to model the entire space of marginal constraints in OT. That is, I don’t think $p(x)\pi(y|x)$  will cover  $\Pi(\mu,\nu)$. Lemma 1 just shows that under some conditions, $p(x)\pi(y|x)$ lies inside $\Pi(\mu,\nu)$, but it would be interesting to see if then entire space of $\Pi(\mu,\nu)$  is covered by the neural contrastive model.
>
> Our response: Related to our response to Concern (1), for $\pi_{forward}(x,y)=p(x)\pi(y|x)$, we have $\int\pi_{forward}(x,y)dy=p(x)$ but allow either $\int\pi_{forward}(x,y)dx =  p(y)$ or $\int\pi_{forward}(x,y)dx \neq p(y)$, whereas for  $\pi\in\Pi(\mu,\nu)$, we have both $\int\pi(x,y)dy=p(x)$ and $\int \pi(x,y)dx= p(y)$. Therefore, the set of all possible joint probability measures, which $\pi_{forward}$ belongs to, covers $\Pi(\mu,\nu)$, in other words, the entire space of $\Pi(\mu,\nu)$ is covered by the space of the joint probability measure of the forward CT. A similar conclusion can also be made for $\pi_{backward}(x,y)=p(y)\pi(x|y)$. We note Lemma 1 adds these conditions, including that $p(x)$ is equal to $p(y)$ in distribution (not required by a regular $\Pi(\mu,\nu)$), to make $\int\pi_{forward}(x,y)dx =  p(y)$ and $\int\pi_{backward}(x,y)dx =  p(x)$.

---

> > ### Comment · AnonReviewer3 · 2020-11-11
> > **Imposing less constraints on joint probability measure can have issues**
> >
> > I understand that from your formulation, marginal constraints are not satisfied and it imposes less constraints on the transportation plan.
> >
> > But this could potentially have some issues.
> >
> > For example, consider a problem where you are required to find OT between two empirical distributions $p(x)$ and $p(y)$ of $n$ samples each. Let $p(x)$ have one sample at $x=-1$, and the rest of the $n-1$ points are drawn from $\mathcal{N}(-1000, 1)$. Similarly, let $p(y)$ have one sample at $y=1$, and the rest of the $n-1$ points are drawn from $\mathcal{N}(1000, 1)$. In this case, what would happen in your formulation is that all samples is $p(x)$ would be transported to $y=1$, and all points in $p(y)$ would be transported to $x=-1$. This could lead to a totally different estimate of OT distance. This is the issue with not satisying $p(x)\pi(y|x) = p(y)\pi(x|y)$.
> >
> > Now, from practical standpoint, this could actually have some issues like missing rare modes in GANs / amplifying biases in a dataset. For instance, if the real distribution has most samples in one category, and a few rare modes, you could tend to miss the rare modes. The reason is because of same arguments I made above.  Identifying these failure cases will be crucial.
> >
> > Out of curiosity, while the conditional transport plan is a lower bound to the true transportation cost, is it possible to talk about the tightness of the lower bound in some way? This would depend on the data distributions too I guess, as I can always construct a distribution where the gap is really high.

---

> > > ### Author Response · Authors · 2020-11-12
> > > **Transport between two distributions p(x) and p(y) is different from that between two empirical distributions $\hat{p}_N(x)$ and $\hat{p}_M(y)$**
> > >
> > > Thank you for your insightful comments. The example you mentioned is indeed a very good one to help further illustrate the properties of the proposed CT and ACT divergence. We are working on adding a paragraph on that example into the revised paper. Before we are able to finish our revision, we'd like to first provide our response to your concern regarding this specific example.
> > >
> > > We'd like to clarify that $$OT(p(x),p(y))\neq E_{x_{1:N}\sim p(x),y_{1:M}\sim p(y)}[OT(\hat p_N(x), \hat p_M(y)].$$
> > > Suppose $p(x)$ is a mixture of a point mass at $x=-1$ and a Gaussian distribution $\mathcal N(-1000,1)$, there is no guarantee that its empirical distribution $\hat p_{N}(x)$, supported on a random mini-batch of $N$ data points $x_{1:N}\sim p(x)$, will have at least one sample at $x=-1$. Similarly, for $p(y)$ that is a mixture of a point mass at $y=1$ and a Gaussian distribution $\mathcal N(1000,1)$, there is no guarantee that its empirical distribution $\hat p_{M}(y)$, supported on $y_{1:M}\sim p(y)$,  will have at least one sample at $y=1$. When $1\notin y_{1:M}$, then obviously, no points in $p(x)$ will be transported to $y=1$ under the ACT divergence. When $1\in y_{1:M}$, the probability for $x\sim p(x)$ to be transported to $y=1$ will be controlled by the navigator parameter and it could be far below 1. Below we provide a concrete example to further clarify this point.
> > >
> > > Consider a special case that $p_X(x)=0.01 \delta_{-1} + 0.99\mathcal N(-1000,1)$ and $p_Y(y)=0.01 \delta_1 + 0.99\mathcal N(1000,1)$ and assume $N=M=64$. Then for an empirical distribution $\hat p_M(y)=\frac{1}{M}\sum_{j=1}^M \delta_{y_j}$, where $y_{1:M}\stackrel{iid}\sim p_Y(y)$, the probability that $1\notin y_{1:M}$ is as large as $0.99^M = 0.99^{64} = 52.56$\%. Therefore, for a random mini-batch, it is true with 52.56\% probability that $1\notin y_{1:M}$, and hence $\hat \pi_M(y=1|x)=0$, which means no samples from $p(x)$ would be transported to $y=1$. When $1\in y_{1:M}$, which happens with probability 47.44\%, the value of $\hat \pi_M(y=1|x)$ will be depending on the navigator parameter $\phi$. For example, setting $d(T_{\phi}(x),T_{\phi}(y)) = \frac{(x-y)^2}{2 e^{\phi}}$, we have
> > > $$
> > > \hat\pi_M(y | x)
> > > \stackrel{def.}=\sum_{j=1}^M \frac{e^{-\frac{(x-y_j)^2}{2e^\phi}}}{\sum_{j'=1}^M e^{- \frac{(x-y_j)^2}{2e^\phi}}} \delta_{y_j},~~y_j\stackrel{iid}\sim p_Y(y);
> > > $$
> > > In this case, suppose $1\in y_{1:M}$, if $e^{\phi}$ is neither too small nor too large, such as $e^{\phi}=1$, then $\hat\pi_M(y =1|x)\approx 1$, but  if $e^{\phi}$ is large, such as $e^{\phi}=e^{100}$, then $\hat\pi_M(y =1|x)\approx\frac{\sum_{j=1}^M \mathbf{1}(y_j=1)}{M}$, which becomes $0.01$ as $M\rightarrow \infty$.

---

> > > > ### Author Response · Authors · 2020-11-12
> > > > **The CT divergence, which does not imply having optimized conditional transport plans, is in general not a lower bound of Wasserstein distance**
> > > >
> > > > "Out of curiosity, while the conditional transport plan is a lower bound to the true transportation cost, is it possible to talk about the tightness of the lower bound in some way? This would depend on the data distributions too I guess, as I can always construct a distribution where the gap is really high."
> > > >
> > > > To address this comment, we'd like to clarify that we are not proposing the CT divergence as an approximation of the Wasserstein distance, and in general, the CT divergence is not smaller than the Wasserstein distance, as explained below.
> > > >
> > > > For conditional transport, the CT divergence is defined as
> > > > $CT(\mu,\nu)  =0.5 CT_{forward}(\mu,\nu)  + 0.5CT_{backward}(\mu,\nu) $, where
> > > > $$ CT_{forward}(\mu,\nu) = E_{(x,y)\sim\pi_{forward}(x,y) }[c(x,y)],~CT_{backward}(\mu,\nu) = E_{(x,y)\sim\pi_{backward}(x,y) }[c(x,y)],$$
> > > > whereas for optimal transport,  the Wasserstein distance is defined as
> > > > $$
> > > > \mathcal W(\mu,\nu) = \min_{\pi\in \Pi(\mu,\nu)} E_{(x,y)\sim\pi(x,y) }[c(x,y)].
> > > > $$
> > > > While it is true that
> > > >  $$\min_{\pi(y|x)} CT_{forward}(\mu,\nu)  \le \mathcal W(\mu,\nu) \text{ and } \min_{\pi(x|y)} CT_{backward}(\mu,\nu)  \le \mathcal W(\mu,\nu) ,$$ it is quite possible that $CT_{forward}(\mu,\nu)>\mathcal W(\mu,\nu)$ and  $CT_{backward}(\mu,\nu)>\mathcal W(\mu,\nu)$ when the conditional transport plans ${\pi(y|x)}$ and ${\pi(x|y)}$ have not been optimized. We note the first subplot of Figure 1 can be used to illustrate this point, as it shows $CT_{forward}(\mu,\nu)$,  $CT_{backward}(\mu,\nu)$, and $CT(\mu,\nu)$ are all above $\mathcal W(\mu,\nu)$ when $p_\theta(y)=\mathcal{N}(0,e^\theta)$ and the navigator parameter $\phi$ have not been optimized, and they all converge towards $\mathcal W_2(\mu,\nu)^2=(1-e^{\theta/2})^2$ when $\theta\rightarrow 0$ (implying $\nu\rightarrow \mu$) and $e^{\phi}\rightarrow 0$. Moreover, Figure 2 shows that $ACT_{forward}(\hat{\mu},\hat{\nu})$, $ACT_{backward}(\hat{\mu},\hat{\nu})$, and $ACT(\hat \mu,\hat \nu)$ are all clearly above $\mathcal W_2(\hat\mu,\hat\nu)^2$ during the initial phase of the training, and then oscillate around (sometimes go below) $\mathcal W_2(\hat\mu,\hat\nu)^2$ when both the generator and navigators have been sufficiently optimized.

---

> ### Author Response · Authors · 2020-11-21
> **Point-by-point response to AnonReviewer3**
>
> Thank you for your constructive comments and insightful questions. Adding to our earlier response, we’d like to provide a comprehensive response to all remaining questions/concerns.
>
> > Q1: Several related comments on OT versus CT/ACT are summarized as follows: “It would be good to ensure the symmetry of the forward and backward transport to satisfy the OT constraints. Imposing fewer constraints on joint probability measure can have issues such as missing modes.”
>
> Response: Compared to f-divergence such as the JS divergence and KL divergence, optimal transport (OT) is often considered more attractive in theory when applied to distribution matching, as it provides useful gradient when the distributions have non-overlapping support. We clarify that CT and ACT, while being related to the Wasserstein distance in its primal form in terms of how the expected transport cost between two distributions is defined, are not designed to serve as an approximation of OT. In fact, both our analysis and experimental results (Fig.3 of the revised paper) suggest that in a mini-batch setting, ACT is preferred to OT due to its amortization mechanism when computing its conditional transport plans using the navigators, whose parameter is updated via SGD across mini-batches.
>
> Thank you for the suggested example, which we have employed to better reveal the difference between OT and CT/ACT and address the corresponding concern:
>
> We agree that OT could help avoid the missing mode issues of standard GANs, however, this is under the assumption that a sufficiently large mini-batch size can always be used. More specifically, if using OT, one often uses the sample Wasserstein distance $\mathcal W(\hat \mu_N,\hat \nu_N)$  between mini-batch based probability measures $\hat \mu_N$ and $\hat \nu_N$ as the loss function. The behavior of the transport plan, which is recomputed for every mini-batch, could vary significantly across different mini-batches, especially if the mini-batch size is not sufficiently large, in which case the learning signal will likely exhibit inconsistency across different training epochs, and this could lead to mode average problems (see Fig. 3 for more details).
>
> To be more specific, suppose we randomly partition the full dataset into non-overlapping mini-batches, compute the transport plan for each mini-batch, and then aggregate the transport plans of all mini-batches to reconstruct the transport plan of the full dataset. This reconstructed transport plan, however, could be significantly different from the true transport plan of the full dataset if the mini-batch size is not sufficiently large. For example, suppose
>
> $X =${$-1001, -1000, -999, -1$ } and $Y = $ { $1, 999, 1000, 1001$ },  then the OT plan will be in an ordered way:
> $$-1001 \leftrightarrow 1;  -1000 \leftrightarrow 999; -999 \leftrightarrow 1000; -1 \leftrightarrow 1001$$
> If we learn the OT plan with mini-batch size 2, like with mini-batches
>
> $X =$ {$-1, -1000$} $\cup$ {$-1001,-999$} and $Y =$ {$1, 999$} $\cup$ {$1000, 1001$}, then the reconstructed OT plan will become
> $$-1001 \leftrightarrow 1000;  -1000 \leftrightarrow 1; -999 \leftrightarrow 1001; -1 \leftrightarrow 999,$$ which is significantly different from the true OT plan between $X$ and $Y$. As in practice, we need to shuffle data in every training epoch, this problem could be severe.
>
> By contrast, the parametric navigator, whose parameter is optimized globally across mini-batches, well amortizes this stochastic effect from mini-batches, allowing the learned conditional transport plans to stay relatively invariant across a wide range of mini-batch sizes (Fig. 9 in the revised version, also implied by Fig. 3). This could be a nice property of CT/ACT compared to OT.
>
>
> > Q2: Concern on the risk of transport plan collapse
>
> Response:  We have carefully studied the risk for the conditional transport plans to degenerate to point mass distributions. (Appendix B.2). In summary, ACT draws mini-batches, consisting of iid samples, to calculate both the forward and backward conditional transport plans and update the navigator parameter with SGD. It could well prevent the conditional transport plans from collapsing into point mass distributions.

---

> > ### Author Response · Authors · 2020-11-21
> > **Point-by-point response to AnonReviewer3 (continue)**
> >
> > > Q3: Is the optimization harder (or at least equally harder) compared to standard GANs? Do you see any optimization benefits compared to standard GANs?
> >
> > Response: First, we note ACT-GAN will degenerate to behave like a standard GAN if we make its conditional transport distributions approach discrete uniform distributions. For example, if we choose $d(\mathcal{T}_\phi (x), \mathcal{T}_\phi(y) ) = \frac{(x-y)^2}{e^\phi} $ and let $\phi \rightarrow \infty$. In this case, the interaction between samples will be flattened and equally treated when we calculate the summary statistics of a mini-batch.
> >
> > Second, adding the navigators into the framework does not make the optimization more difficult to handle. On the contrary, amortizing the computation of the conditional transport plans, it becomes robust to the mini-batch size, provides strong resistance to model collapse, and consistently helps achieve better performance (see the comparisons of ACT with standard GANs in Figs. 4, 10-15). In our ablation study (Fig. 19), we show ACT allows alternative loss (e.g., standard cross-entropy discriminator loss) to train the critic, which further demonstrates its robustness.
> >
> > > Q4: “The results of image-based generative modeling is not that impressive. On CIFAR and LSUN datasets, the performance is similar / slightly better than the compared GAN models. Also, many SOTA GAN models are not compared. So, it is very hard to say if the proposed model advances SOTA results.”
> >
> > Response: We’d like to clarify that the purpose of this paper is not to propose a better GAN architecture (such as BigGAN) or training strategy (such as progressive growing, using a very large mini-batch size, applying the truncation trick for the input noise, and adding data augmentation), both are very important to achieve SOTA results in addition to the choice of the statistical distance between two distributions. Considering these SOTA GANs often consist of additional components in the generator/critic or need additional highly-specialized techniques/heuristics during training/generation, it will not be simple to perform the ablation study to attribute the performance boost to ACT. In addition, these SOTA GANs often require intensive GPU computing power to train, which is not cheap at all to afford. Therefore, we have selected two representative backbones (DCGAN and SNGAN), which we could afford to train, and just substituted their loss with ACT. We can observe that compared to their previous performance, ACT boosts the model and shows large improvements.
> >
> > In the revision, we have also evaluated ACT with higher-resolution images to see its performance. The generation quality assures that ACT is able to handle more complex visual tasks (see Fig. 7). We are working on applying ACT to more advanced models, and believe the significant improvement over SNGAN can be further extended with more advanced structure change (such as BigGAN) and a more sophisticated training strategy (such as progressive growing and adding the truncation trick to the input noise).
> >
> > > Q5: “I would have liked to see more interesting results. The formulation of authors gives an estimate of transportation plan $\pi(x,y)$ in addition to the generative model itself, which is not possible to estimate in dual-based OT GAN optimization.The transportation plan can be used in interesting applications. One possibility is to estimate likelihoods and possibly use in OOD detection. ”
> >
> > Response: We appreciate your sharing the idea of using the estimated conditional transport plan in OOD detection. We have added Balaji et al. into our reference and we will investigate that idea in our future study.
> >
> > As measuring the distance between two probability distributions is a fundamental problem, we believe there are many potential tasks to tackle with the CT/ACT divergence. Indeed, we have already started working on applying ACT’s conditional transport plans to a wide variety of tasks, including contrastive representation learning, image-to-image translation, and imitation learning (inverse reinforcement learning) tasks, to name a few. We are excited about our preliminary results on these tasks, but discussing them or OOD detection is beyond the scope of this paper.

---

### Author Response · Authors · 2020-11-20
**Summary of major revisions**

Dear Reviewers,

Thank you for your insightful and constructive feedback. In the revised manuscript, we have carefully addressed all major concerns and provided a number of additional experiments to help answer your questions. Our major changes, highlighted in blue in the revised manuscript, are summarized as follows:

1. We have clarified the role of the navigators in “amortizing” the computation of the conditional transport plans, which helps avoid the need for an iterative procedure for each mini-batch as required to compute the sample Wasserstein distance in its primal form as well as the Sinkhorn distance (Page 4).

2. We have clarified why the Wasserstein distance often suffers from biased gradients (Page 3 for the dual form and Page 4 for the primal form).

3. We have added experiments to demonstrate that SGD optimization under the ACT divergence is robust to the mini-batch size, while that under several other different distance measures, including the sample Wasserstein distance, is not (Figs. 3 and 4 in the revised paper).

4. We have demonstrated that a generator trained under the ACT has strong resistance to mode collapse (Fig. 4 in the revised paper).

5. In particular, our newly added results suggest that using only the forward ACT cost encourages the generator to cover the modes while using only the backward ACT cost encourages it to seek a mode (Figs. 1, 5, 8 in the revised paper). This explains why the ACT, which equally combines the forward and backward costs, strikes a good balance between mode covering and seeking behaviors and consequently builds strong resistance to mode collapse.

6. To address the concern of Reviewer3, we have carefully studied the risk for the conditional transport plans to degenerate (Appendix B.2).

7. We have added additional example generation results on higher-resolution images (Fig. 7 of the revised paper)

Below please find a separate point-by-point response to each reviewer’s comments.

---

### Decision · Program_Chairs · 2021-01-07
**Final Decision**

**Decision:**

Reject

**Comment:**

The paper proposes a new measure of difference between two distributions using conditional transport. The paper considers an important problem.  However, some major concerns remain after the discussion among the reviewers. In particular, the paper focuses on the evaluation on a toy dataset. It is unclear whether the claim carries over to large real datasets. The presentation of the paper also needs substantial improvement.